# Integrable fishnet circuits and Brownian solitons

Žiga Krajnik[1*], Enej Ilievski[2], Tomaž Prosen[2,3], Benjamin J. A. Héry[4] and Vincent Pasquier[5]

**1** Department of Physics, New York University, 726 Broadway, New York, New York 10003, USA
**2** Department of Physics, Faculty of Mathematics and Physics, University of Ljubljana, Jadranska 19, SI-1000 Ljubljana, Slovenia
**3** Institute of Mathematics, Physics and Mechanics, Jadranska 19, SI-1000, Ljubljana, Slovenia
**4** Fachbereich Physik, Freie Universität Berlin, 14195 Berlin, Germany
**5** Institut de Physique Théorique, Université Paris Saclay, CEA, CNRS UMR 3681, 91191 Gif-sur-Yvette, France

⋆ ziga.krajnik@nyu.edu

November 13, 2024

## Abstract

**We introduce classical many-body dynamics on a one-dimensional lattice comprising local two-body maps arranged on discrete space-time mesh that serve as discretizations of Hamiltonian dynamics with arbitrarily time-varying coupling constants. Time evolution is generated by passing an auxiliary degree of freedom along the lattice, resulting in a 'fishnet' circuit structure. We construct integrable circuits consisting of Yang-Baxter maps and demonstrate their general properties, using the Toda and anisotropic Landau-Lifschitz models as examples. Upon stochastically rescaling time, the dynamics is dominated by fluctuations and we observe solitons undergoing Brownian motion.**

# 1   Introduction

Exactly solvable models play a crucial role in our understanding of diverse physical systems by serving as paradigms of the underlying phenomena. Their importance is further sharpened in the realm of many-body physics where few general analytical tools are available and even numerical simulations of large systems quickly become computationally intensive.

The approach of statistical physics is to shift the focus from following all the microscopic degrees of freedom, replacing them by a coarse-grained description in terms of mesoscopic averages, while the remnants of the microscopic degrees of freedom often enter the description stochastically. The prototypical example of this approach is Brownian motion, describing a particle diffusing due to fluctuations of its environment which is modeled as stochastic noise. While averages are easier to describe due to concentration of measure, the approach often entails approximation which are difficult to control, especially in strongly-interacting systems.

An alternative approach, feasible only for a small subset of systems, is to first solve the initial value problem exactly and average over an ensemble of initial conditions only afterwards. Many of these models owe their solubility to integrability, manifested as the existence of extensively many conserved quantities which stabilize solitons - stable nonlinear excitations that preserve their identity upon scattering. In classical systems the conservation laws constraint the dynamics sufficiently to allow for a complete separation of a many-body system in terms of canonical variables, as exemplified by the inverse scattering transformation [1–3], or finite gap integration [4] which solve the initial value problem for an integrable many-body system. On the quantum side the Bethe ansatz [5–7] reduces the problem of diagonalizing an exponentially large matrix, an daunting task for all but the smallest systems, to the comparatively much simpler problem of solving a system of algebraic equations.

Despite these powerful analytical techniques, the investigation of dynamical properties of ensembles of initial conditions remains a challenging task. An important step forward in this direction has been the development of generalized hydrodynamics, a hydrodynamic theory for integrable systems [8, 9]. However, at present the assumptions of the theory limit its applicability to leading-order dynamics on the ballistic scale. To access dynamics on sub-ballistic scales a recourse to numerical simulations remains a necessity.

However, numerical simulations of integrable systems hide potential pitfalls as conventional discretizations invariably break many conservation laws, spoiling the asymptotic properties of the discretization. This problem has been addressed by the development of explicit space-time discretizations which take the form of 'brickwork' circuits [10] which are integrable by construction.

While integrable brickwork discretizations have been successfully applied to numerically study dynamical properties of both classical [11–13] and quantum systems [14–17] and even implemented in modern quantum computing platforms [18–21], the approach has two draw-

backs. Firstly, each layer of the circuit must have the same time step to avoid breaking integrability. Secondly, the conserved quantities of the circuit are deformed by the time-step and converge to their continuous counterparts only in the limit of a large number of layers.

## 1.1 Summary of results

In this paper, we construct a family of integrable classical dynamical systems in discrete space-time, representing integrable discretizations of Hamiltonian systems. Our construction employs space-time lattices of 'fishnet' type, unlike the commonly used 'brickwork' circuits. Circuits of similar type arise in the quantum transfer matrix formalism [22], where they correspond to Trotter approximations of Gibbs states in integrable quantum spin chains. Here, the circuits instead represent a canonical dynamical map acting on a many-body classical phase space. By utilizing Bäcklund transformations [23–25], we are able to fabricate integrable circuits that depend on an arbitrary number of time-step parameters, which allows us to interpret them as integrability preserving space-time discretizations of integrable Hamiltonian flows with time-dependent coupling constants. The constructed circuits preserve the conservation laws of the continuous-time Hamiltonian dynamics. By regarding the time-step parameters as random variables, we obtain an integrable stochastic dynamics in discrete space-time. We highlight several prominent physical features of such stochastic integrable fishnet circuits. Particularly, we exhibit Brownian solitons and study the corresponding dressed dynamical structure factors.

The central algebraic relation of our construction is a matrix refactorization problem [26–28] of the form,

$$\mathcal{L}_{\lambda^-}^a(y')\mathcal{L}_{\lambda^+}(x') = \mathcal{L}_{\lambda^+}(x)\mathcal{L}_{\lambda^-}^a(y), \tag{1.1}$$

for a pair of matrices $\mathcal{L}(x)$ and $\mathcal{L}^a(y)$ depending on a spectral parameter $\lambda \in \mathbb{C}$, with the shifted parameters $\lambda^\pm \equiv \lambda \pm \tau/2$ and a free time-step parameter $\tau \in \mathbb{R}$. Here, variables $x$ and $y$ represent two local classical degrees of freedom. Given a pair $(x, y)$ as an input, the solution to Eq. (1.1) defines a one-parameter family of 'dynamical maps' $\psi_\tau$ from $(x, y) \mapsto (x', y')$, see Figure 1.

In the realm of integrable systems, equation of the form (1.1) plays a role of a discrete zero-curvature conditions [3], ensuring compatibility of the auxiliary linear transport problem. In this view, we shall refer to $\mathcal{L}$ as the physical Lax matrix, while the superscript 'a' indicates that $\mathcal{L}^a$ is the auxiliary Lax matrix. Correspondingly, the variable $x$ is interpreted as the physical local degree of freedom of the model, while $y$ is associated with the auxiliary degree of freedom. We proceed by introducing the monodromy matrix of length $L$,

$$\mathcal{M}_\lambda(X) = \mathcal{L}_\lambda(x_L)\dots\mathcal{L}_\lambda(x_2)\mathcal{L}_\lambda(x_1), \tag{1.2}$$

depending on the set of physical variables $X = (x_1, x_2, \dots, x_L)$. Given $y_0$ as an input, an iterative application of the local relation (1.1) yields the relation

$$\mathcal{L}_{\lambda^-}^a(y_L)\mathcal{M}_{\lambda^+}(X') = \mathcal{M}_{\lambda^+}(X)\mathcal{L}_{\lambda^-}^a(y_0), \tag{1.3}$$

depicted in Figure 2, where $y_L$ in the final (output) $y$ variable. We assume that there exists an initial $y_0$ such that $y_L = y_0$, thereby closing the chain periodically. The outcome is a chiral many-body left propagator, denoted hereafter by $\Phi_\tau^-$, that maps $X \mapsto X'$ in the vertical (i.e. time) direction.

We also introduce the right propagator $\Phi_\tau^+$, being simply the inverse of the left chiral propagator $\Phi_\tau^-$ with the same time-step,

$$\Phi_\tau^\pm \circ \Phi_\tau^\mp = \text{id}, \tag{1.4}$$

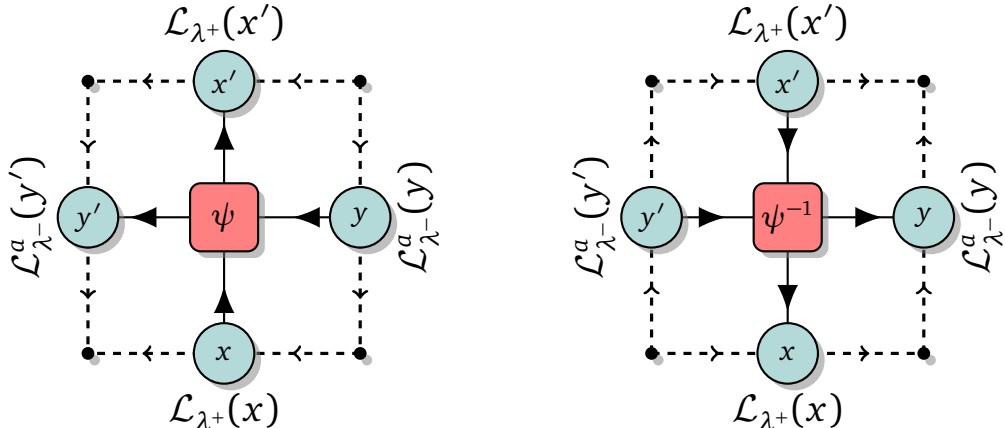

Figure 1: (left panel) A two-body map $\psi$ (red square, time-step not shown) located at the middle of the square plaquette maps the input edge variables $x$ and $y$ (green circles) from the bottom and right sides of the plaquette to the output edge variables $x'$ and $y'$ on the top and left sides. Oriented dashed black lines indicate linear transport of auxiliary vertex variables (not shown) induced by physical (horizontal direction) and auxiliary (vertical direction) Lax matrices. (right panel) Reversing the direction of propagation yields the inverse two-body map $\psi^{-1}$ which maps the variables $x'$ and $y'$ from the top and left sides to $x$ and $y$ from the bottom and left sides of the plaquette.

where $\circ$ denotes the composition of maps. The right propagator can be realized by interchanging $X$ and $X'$ in Eq. (1.3) which amounts to reversing the arrows in Figure 2. It is important to emphasize that the existence of auxiliary variables $y_0$ and $y_L$ that allowed for the periodic closure are crucial for the well-posedness of the defined propagators. Multiplying by the inverse of $\mathcal{L}^a$ and taking the trace of Eq. (1.3), periodicity in the spatial (i.e. horizontal) direction indicates that the time-propagators $\Phi_\tau^\pm$ conserve the transfer functions $\mathcal{T}_\lambda^{(n)} = \operatorname{Tr} \mathcal{M}_\lambda^n$

$$\mathcal{T}_\lambda^{(n)} \circ \Phi_\tau^\pm = \mathcal{T}_\lambda^{(n)}. \tag{1.5}$$

The time-evolution of the monodromy matrix is thus isospectral, and we can use the transfer functions $\mathcal{T}_\lambda^n$ as generating functions of local conserved quantities in involution.
A detailed construction of the time-propagators is given in Section 2. There we show that, upon imposing the periodic closure, canonicity of the local two-body map $\psi$ gets naturally extended to full propagators $\Phi_\tau^\pm$. Moreover, using the fact that $\psi$ is a Yang-Baxter map, we establish commutativity of the chiral time-propagators for arbitrary time-step parameters

$$\Phi_\tau^+ \circ \Phi_{\tau'}^\pm = \Phi_{\tau'}^\pm \circ \Phi_\tau^+. \tag{1.6}$$

This type of commutation property will play a pivotal role in our subsequent study of dynamical properties. Specifically, it will permit us to randomly sample the time-step parameters upon iterating the elementary time propagators without breaking integrability. Hence, our construction is fundamentally different from the more standard 'brickwork' type discretization of integrable dynamics [10–13].

In Section 3 we introduce a composite propagator $\Phi_\tau$ by composing both chiral propagators,

$$\Phi_\tau = \Phi_{-\tau}^- \circ \Phi_\tau^+. \tag{1.7}$$

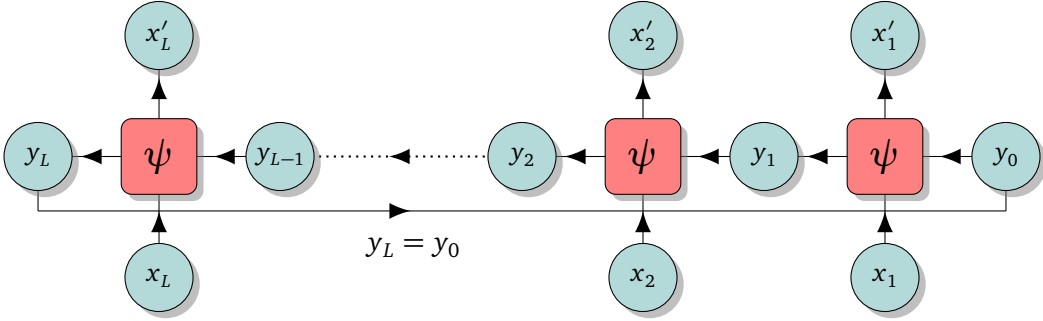

Figure 2: A chain of $\psi$-maps, representing propagation of an auxiliary initial variable $y_0$ (right green circle) from right to left (left green circle) subject to the periodic closure, $y_0 = y_L$, realizes the left chiral propagator $\Phi^-$ that maps physical input variables $X$ (bottom green circles) to the output variables $X'$ (top green circles) respectively. The right chiral propagator $\Phi^+$ is obtained by reversing the directions of arrows.

The full many-body evolution for time $t \in \mathbb{N}$ is then simply given by a sequence of composite propagators $\Phi_\tau$, namely

$$\Phi_\tau^{(t)} = \Phi_{\tau_t} \circ \ldots \circ \Phi_{\tau_2} \circ \Phi_{\tau_1}, \tag{1.8}$$

depending on an arbitrary sequence of time-step parameters $\tau = (\tau_1, \tau_2, \ldots, \tau_t)$, see Figure 3. In addition, we establish integrability of (1.8) and introduce a family of invariant measures which we later use in our applications.

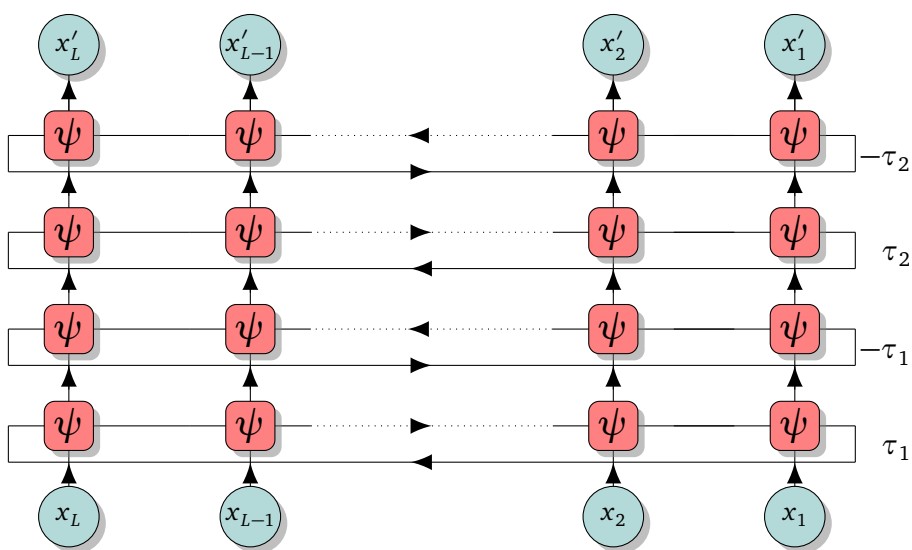

Figure 3: The finite-time propagator $\Phi_\tau^{(t)}$ (shown for $t = 2$) is given by a sequential application of $t$ composite propagators $\Phi_\tau$ (1.7) depending on arbitrary time-steps $\tau_j$. Composite propagators $\Phi_\tau$ are composed from the right (odd layers) and left (even layers) elementary chiral propagators $\Phi_{\mp\tau}^\pm$ related by the inversion relation (2.39) via the corresponding Lax matrices. Propagation of auxiliary variables in the horizontal direction is suppressed for clarity.

We proceed in Section 4 by discussing particular examples. Focusing on the classical Toda lattice and the anisotropic lattice Landau-Lifshitz spin chain, we explicitly construct their respective local maps with corresponding fixed points and discuss several other key properties.

These examples serve to elucidate the preceding algebraic construction. We also give the continuous-time limit of the discrete-time dynamics.

Finally, in Section 5 we numerically investigate the dynamical behavior of our circuits. We mainly focus on the stochastic version obtained by averaging the evolution over time sequences $\tau$ sampled from a given probability distribution $\Omega$, that is

$$\overline{\Phi}_\Omega^{(t)} = \int d\tau_1 d\tau_2 \dots d\tau_t \, \Omega(\tau) \Phi_\tau^{(t)}. \tag{1.9}$$

As a simple example, we consider statistically independent time-steps, $\Omega(\tau) = \prod_{j=1}^t \omega_2(\tau_j)$, sampled from a discrete two-point probability distribution

$$\omega_2(u) = \frac{1-\eta}{2}\delta(u+\tau) + \frac{1+\eta}{2}\delta(u-\tau), \qquad -1 \le \eta \le 1. \tag{1.10}$$

When $\eta \ne 0$, the distribution's mean is non-zero, and we find that the stochastic dynamics, shown in Figure 4, becomes asymptotically deterministic with an effective rescaled timescale,

$$\overline{\Phi}_{\omega_2}^{(t)} \simeq \Phi_\tau^{(\lfloor t\eta \rfloor)}. \tag{1.11}$$

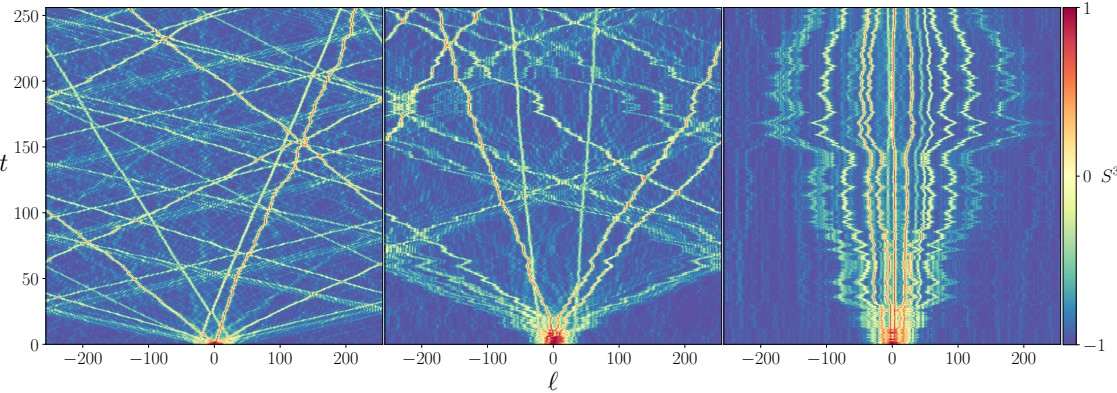

Figure 4: Time evolution fo the spin profile $S^3 = S^3(x,t)$ from from an initial Gaussian profile, corresponding to the easy-plane regime of the anisotropic Landau-Lifshitz model with time-steps sampled from the two-point measure $\omega_2$ (1.10): (left panel, $\eta = 1$) deterministic dynamics featuring ballistic soliton trajectories; (middle panel, $\eta = 1/2$) stochastic microscopic dynamics governed by deterministic dynamics at leading order indicated by asymptotically ballistic soliton trajectories, see Eq. (1.11); (right panel, $\eta = 0$) subballistic spreading of asymptotic trajectories with diffusive broadening governed by the effective stochastic hydrodynamics on the scale $t^{1/2}$, see Eq. (1.12).

On the other hand, the mean vanishes for $\eta = 0$ and the stochastic dynamics is now governed by fluctuations that take place on the square-root scale. The latter are asymptotically normal and therefore the effective evolution reads

$$\overline{\Phi}_{\omega_2}^{(t)} \simeq \int_{\mathbb{R}} \frac{du}{\sqrt{2\pi}} \, e^{-u^2/2} \, \Phi_\tau^{(\lfloor ut^{1/2} \rfloor)}. \tag{1.12}$$

By invoking the central limit theorem, we show that such stochastic dynamics with independent and identically distributed time-steps is universal in the continuous-time limit for any probability distribution with a finite second moment.

Finally, we characterize the dynamical structure factor of the ensuing stochastic dynamics featuring 'Brownian solitons'. Specifically, by means of a dressing transformation, we express it in terms of the the structure factor of the underlying deterministic dynamics.

## 2 Elementary chiral propagators

In this section we provide the algebraic construction of the elementary chiral propagators.

**Phase spaces and Poisson structure**    Integrable fishnet circuits involve physical and auxiliary classical degrees of freedom. To this end, let $\mathcal{X}$ and $\mathcal{Y}$ denote the local physical and auxiliary phase spaces, representing symplectic manifolds of dimensions $\dim \mathcal{X} = 2n_{\mathcal{X}}$ and $\dim \mathcal{Y} = 2n_{\mathcal{Y}}$, respectively. In general, we do no assume the phase spaces $\mathcal{X}$ and $\mathcal{Y}$ to be isomorphic.

To equip $\mathcal{X}$ and $\mathcal{Y}$ with Poisson brackets, we utilize Lax matrices, representing matrix-valued functions on their respective Poisson manifolds. We introduce the 'physical' and 'auxiliary' Lax matrices, $\mathcal{L}(\lambda, x) : \mathbb{C} \times \mathcal{X} \to \mathbb{C}^{n \times n}$ with $x \in \mathcal{X}$ and $\mathcal{L}^a(\lambda, y) : \mathbb{C} \times \mathcal{Y} \to \mathbb{C}^{n \times n}$ with $y \in \mathcal{Y}$, respectively, depending on the analytic spectral parameter $\lambda \in \mathbb{C}$ and respective local degrees of freedom. We assume both Lax matrices to be invertible. For compactness of notation, we regard $\lambda$ as a parameter and denote $\mathcal{L}_\lambda(x) \equiv \mathcal{L}(\lambda, x)$ so that $\mathcal{L}_\lambda : \mathcal{X} \to \mathbb{C}^{n \times n}$ and likewise for $\mathcal{L}^a_\lambda$.

The Poisson brackets on $\mathcal{X}$ can be specified in terms of the quadratic Sklyanin bracket,

$$\left\{ \mathcal{L}_\lambda(x_\ell) \overset{\otimes}{,} \mathcal{L}_\mu(x_{\ell'}) \right\} = \left[ r_{\lambda - \mu}, \mathcal{L}_\lambda(x_\ell) \otimes \mathcal{L}_\mu(x_{\ell'}) \right] \delta_{\ell, \ell'} \tag{2.1}$$

where the so-called classical $r$-matrix [3], $r_\lambda \in \mathrm{End}(\mathbb{C}^n \otimes \mathbb{C}^n)$, plays the role of structure constants. Importantly, the Poisson brackets on $\mathcal{Y}$ can similarly be realized with the same $r$-matrix upon replacing $\mathcal{L}_\lambda$ with $\mathcal{L}^a_\lambda$.[1] This will ensure mutual compatibility of both Lax matrices.

**Discrete zero-curvature condition**    With a pair of Lax matrices in hand, we consider the zero-curvature condition in discrete space-time, taking the form of a local exchange relation,

$$\mathcal{L}^a_{\lambda^-}(y') \mathcal{L}_{\lambda^+}(x') = \mathcal{L}_{\lambda^+}(x) \mathcal{L}^a_{\lambda^-}(y), \tag{2.2}$$

where $\lambda^\pm = \lambda \pm \tau/2$ and $\tau \in \mathbb{R}$. Eq. (2.2) can be viewed as a matrix refactorization problem [26, 27]. We shall assume that it admits a unique solution. In such a case, the solution can be interpreted as a dynamical two-body map $\psi_\tau : \mathcal{X} \times \mathcal{Y} \to \mathcal{X} \times \mathcal{Y}$, i.e.

$$(x', y') = \psi_\tau(x, y), \tag{2.3}$$

with the time-step parameter $\tau$, see Figure 1 for a graphical representation. By reversing the direction of propagation in Eq. (2.2) we similarly define the inverse local propagator

$$(x, y) = \psi_\tau^{-1}(x', y'), \tag{2.4}$$

see right panel of Figure 1.

---

[1] By analogy with quantum $R$-matrices, classical $r$-matrices obeying the classical Yang–Baxter equation do not admit a unique Poisson algebra.

**Monodromy matrix and transfer functions** We next consider a physical chain of length $L \in \mathbb{N}$ with local physical degrees of freedom $X \equiv (x_1, x_2, \ldots, x_L) \in \mathcal{X}^L$, and define an inhomogeneous monodromy matrix $\mathcal{M}_\lambda : \mathcal{X}^L \to \mathbb{C}^{n \times n}$ of length $L$ with inhomogeneities $\Xi = (\xi_1, \xi_2, \ldots, \xi_L) \in \mathbb{R}^L$ as a spatially ordered matrix product of physical Lax matrices,

$$\mathcal{M}_{\lambda,\Xi}(X) = \mathcal{L}_{\lambda-\xi_L}(x_L) \ldots \mathcal{L}_{\lambda-\xi_2}(x_2) \mathcal{L}_{\lambda-\xi_1}(x_1). \tag{2.5}$$

In the following, we mostly suppress the dependence of functions on inhomogeneities $\Xi$ for convenience. By taking traces of the $n$-th powers of the monodromy matrix, we define a family of transfer functions $\mathcal{T}_\lambda^{(n)} : \mathcal{X}^L \to \mathbb{C}$,

$$\mathcal{T}_\lambda^{(n)} = \operatorname{Tr} \mathcal{M}_\lambda^n. \tag{2.6}$$

We now construct a pair of chiral propagators $\Phi_\tau^\pm$ that leave the functions $\mathcal{T}_\lambda^{(n)}$ invariant.

**Chiral one-step propagators** The first step is to derive the chiral one-step propagators $\Phi_\tau^\pm$. This is achieved by extending the local exchange relation (2.2) to the monodromy matrix $\mathcal{M}_\lambda$,

$$\mathcal{L}_{\lambda^-}^a(y_L) \mathcal{M}_{\lambda^+}(X') = \mathcal{M}_{\lambda^+}(X) \mathcal{L}_{\lambda^-}^a(y_0). \tag{2.7}$$

One may interpret this as bringing the auxiliary Lax matrix $\mathcal{L}^a$ through the monodromy matrix by starting from the right with an initial value $y_0$. The relation (2.7) is resolved iteratively by repeated applications of the local exchange relation (2.2),

$$(x_1', \ldots, x_\ell', y_\ell) = \Psi_\tau(x_1, \ldots, x_\ell, y_0) = \psi_{\tau-\xi_\ell}(x_\ell, y_{\ell-1}) \circ \Psi_\tau(x_1, \ldots, x_{\ell-1}, y_0), \tag{2.8}$$

as shown graphically in Figure 2. Iteration of the right-hand side of Eq. (2.8) for $\ell = 1, 2, \ldots, L$ with the initial condition

$$\Psi_\tau(y_0) = y_0, \tag{2.9}$$

yields the map $\Psi_\tau : \mathcal{X}^L \times \mathcal{Y} \to \mathcal{X}^L \times \mathcal{Y}$. We assume that $\Psi_\tau$ has a fixed point, denoted by $y_\tau^*$

$$y_\tau^*(X) = \pi_{\mathcal{Y}} \circ \Psi_\tau(X, y_\tau^*(X)), \tag{2.10}$$

where $\pi_{\mathcal{Y}}(X, y) = y$ is a projection onto the auxiliary space $\mathcal{Y}$. By evaluating $\Psi_\tau$ at the fixed point value $y_\tau^*$ we obtain the left one-step chiral propagator $\Phi_\tau^- : \mathcal{X}^L \to \mathcal{X}^L$,

$$\Phi_\tau^-(X) = \pi_{\mathcal{X}} \circ \Psi_\tau(X, y_\tau^*(X)), \tag{2.11}$$

where $\pi_{\mathcal{X}}(X, y) = X$ is a projection onto the physical phase space $\mathcal{X}^L$. Correspondingly, the right one-step chiral propagator $\Phi_\tau^+ : \mathcal{X}^L \to \mathcal{X}^L$ is given by the inverse of the left chiral propagator,

$$\Phi_\tau^\pm \circ \Phi_\tau^\mp = \mathrm{id}. \tag{2.12}$$

The inverse of $\Phi_\tau^-$ is found by swapping $X$ and $X'$ in Eq. (2.7), yielding

$$\mathcal{L}_{\lambda^-}^a(y_L) \mathcal{M}_{\lambda^+}(X) = \mathcal{M}_{\lambda^+}(X') \mathcal{L}_{\lambda^-}^a(y_0), \tag{2.13}$$

which is subsequently resolved by the following iteration

$$(x_\ell', \ldots, x_L', y_{\ell-1}) = \Psi_\tau^{-1}(x_\ell, \ldots, x_L, y_L) = \psi_{\tau-\xi_\ell}^{-1}(x_\ell, y_\ell) \circ \Psi_\tau^{-1}(x_{\ell+1}, \ldots, x_L, y_L), \tag{2.14}$$

and by using Eq. (2.4) to invert $\psi$ for $\ell = L, L-1, \ldots, 1$, with the initial condition

$$\Psi_\tau^{-1}(y_L) = y_L. \tag{2.15}$$

By construction the map $\Psi_\tau^{-1} : \mathcal{X}^L \times \mathcal{Y} \to \mathcal{X}^L \times \mathcal{Y}$ is the inverse of $\Psi_\tau$,

$$\Psi_\tau^{-1} \circ \Psi_\tau = \mathrm{id}, \tag{2.16}$$

and therefore has the same fixed point

$$y_\tau^*(X) = \pi_\mathcal{Y} \circ \Psi_\tau^{-1}(\Phi_\tau^-(X), y_\tau^*(X)), \tag{2.17}$$

Changing variables as $\Phi_\tau^-(X) \to X$, we then have an explicit description of the the left one-step chiral propagator $\Phi_\tau^+$ in the form of

$$\Phi_\tau^+(X) = \pi_\mathcal{X} \circ \Psi_\tau^{-1}(X, y_\tau^*([\Phi_\tau^-]^{-1}(X))). \tag{2.18}$$

Note that the chiral propagators $\Phi_\tau^\pm$ are implicitly defined by the fixed point $y_\tau^*$. When there exist multiple fixed points, each defines a distinct propagator. For simplicity we suppress the propagators' dependence on the choice of fixed points in our notation.

**Conservation laws**    Substituting the fixed point $y_\tau^*$ in place of $y_0$ and $y_L$ on both sides of Eq. (2.7), it is straightforward to show that $\mathcal{M}_\lambda^n(X)$ is similar to $\mathcal{M}_\lambda^n(X')$. The transfer functions $\mathcal{T}_\lambda^{(n)}$ are therefore preserved by the time-evolution with the chiral propagators $\Phi_\tau^\pm$,

$$\mathcal{T}_\lambda^{(n)} \circ \Phi_\tau^\pm = \mathcal{T}_\lambda^{(n)}. \tag{2.19}$$

This shows that the chiral propagators $\Phi_\tau^\pm$ conserve the entire spectral curve

$$\det\left[\mu \mathbb{1} - \mathcal{M}_\lambda \circ \Phi_\tau^\pm\right] = \det\left[\mu \mathbb{1} - \mathcal{M}_\lambda\right]. \tag{2.20}$$

As a direct corollary of (2.1), powers of the monodromy matrix (2.5) satisfy the Sklyanin brackets

$$\left\{\mathcal{M}_\lambda^n \overset{\otimes}{,} \mathcal{M}_\mu^m\right\} = \left[r_{\lambda-\mu}, \mathcal{M}_\lambda^n \otimes \mathcal{M}_\mu^m\right]. \tag{2.21}$$

By tracing out the matrix space, Eq. (2.21) implies Poisson commutativity of transfer functions

$$\{\mathcal{T}_\lambda^{(n)}, \mathcal{T}_\mu^{(m)}\} = 0, \tag{2.22}$$

for arbitrary $n$ and $m$ and spectral parameters $\lambda, \mu \in \mathbb{C}$. By expanding the transfer functions $\mathcal{T}_\lambda^{(n)}$ as power series in $1/\lambda$,

$$\mathcal{T}_\lambda^{(n)} = \sum_{k=1}^{L} Q_k^{(n)} \lambda^{-k}, \tag{2.23}$$

we generate multiple towers[2] (labelled by $n$) of conservation laws $Q_k^{(n)} : \mathcal{X}^L \to \mathbb{C}$ in involution (due to Eq. (2.22)),

$$\left\{Q_k^{(n)}, Q_{k'}^{(n')}\right\} = 0, \tag{2.24}$$

invariant under the time evolution generated by the one-step chiral propagators, i.e.

$$Q_k^{(n)} \circ \Phi_\tau^\pm = Q_k^{(n)}. \tag{2.25}$$

It is worth noting that the conserved quantities (2.23) not do depend on the parameter $\tau$, in stark contrast with the conserved quantities of integrable brickwork circuits [10–13] generated from inhomogeneous commuting transfer functions with staggered spectral parameters. In fact, $Q_k^{(n)}$ are precisely the conserved quantities associated with integrable Hamiltonian flows [3] associated to the physical Lax matrix $\mathcal{L}_\lambda$. In general, the charges $Q_k^{(n)}$ are neither real nor local. Locality typically arises due to extra model-specific symmetry properties of the Lax matrix, see Section 4 and e.g. Refs. [3, 5] and Appendix C of Ref. [13].

---

[2]The total number of functionally independent charges mutually in involution acting in $\mathcal{X}^L$ can be at most $n_\mathcal{X}$.

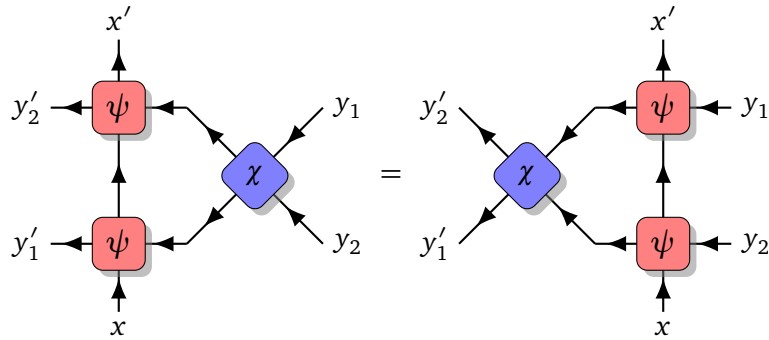

Figure 5: A pair of $\psi$-maps (red squares) and $\chi$ (blue squares) satisfy the Yang-Baxter relation (2.27) (as a consequence of Eq. (2.26) and uniqueness of factorization), an equality between two distinct sequences of transformations between the initial variables $(x, y_1, y_2)$ and the final variables $(x', y_1', y_2')$. Spectral parameters not shown for clarity.

**Yang-Baxter relation**  Given a triplet of initial variables $(x, y_2, y_1)$ and the corresponding product of Lax matrices $\mathcal{L}_\eta(x)\mathcal{L}_\lambda^a(y_2)\mathcal{L}_\mu^a(y_1)$, the local exchange relation (2.2) allows us to reverse the order of factors, yielding $\mathcal{L}_\mu^a(y_1')\mathcal{L}_\lambda^a(y_2')\mathcal{L}_\eta(x')$ depending on the output variables $(x', y_2', y_1')$. This can be achieved in two (apriori inequivalent) ways: either by first interchanging the two $\mathcal{L}^a$ followed by two exchanges involving $\mathcal{L}$, or the other way around. The two exchange schemes are illustrated in Figure 5. Either of these two protocols results in

$$\mathcal{L}_\mu^a(y_1')\mathcal{L}_\lambda^a(y_2')\mathcal{L}_\eta(x') = \mathcal{L}_\eta(x)\mathcal{L}_\lambda^a(y_2)\mathcal{L}_\mu^a(y_1), \tag{2.26}$$

albeit with in principle different output tuples $(x', y_2', y_1')$. Nevertheless, provided that factorization into Lax matrices is unique (see e.g. Ref. [12] for an explicit example), both exchange protocols yield identical results. [3] As a consequence, we deduce a compatibility condition on $\mathcal{X} \times \mathcal{Y} \times \mathcal{Y}$ reading

$$\psi_{\eta-\lambda}^{32} \circ \psi_{\eta-\mu}^{31} \circ \chi_{\lambda-\mu}^{21} = \chi_{\lambda-\mu}^{21} \circ \psi_{\eta-\mu}^{31} \circ \psi_{\eta-\lambda}^{32}. \tag{2.27}$$

The superscripts here indicate the pair of spaces on which the maps acts non-trivially, i.e.

$$f^{21}(x, y_2, y_1) = (x, y_2', y_1'), \tag{2.28}$$

$$f^{32}(x, y_2, y_1) = (x', y_2', y_1), \tag{2.29}$$

$$f^{31}(x, y_2, y_1) = (x', y_2, y_1'). \tag{2.30}$$

The new intertwiner[4] $\chi_\lambda : \mathcal{Y} \times \mathcal{Y} \to \mathcal{Y} \times \mathcal{Y}$ entering Eq. (2.27) is defined as the solution to the local exchange relation involving two auxiliary Lax operator, cf. Eq. (2.2),

$$\mathcal{L}_\mu^a(y_1')\mathcal{L}_\lambda^a(y_2') = \mathcal{L}_\lambda^a(y_2)\mathcal{L}_\mu^a(y_1). \tag{2.31}$$

We note that Eq. (2.27) is the set-theoretic Yang-Baxter equation [29–31].

**Commuting chiral propagators**  The left and right elementary chiral propagators $\Phi_\tau^\pm$ with identical inhomogeneities commute for arbitrary time-step parameters $\tau, \tau' \in \mathbb{R}$, i.e.

$$\Phi_\tau^+ \circ \Phi_{\tau'}^\pm = \Phi_{\tau'}^\pm \circ \Phi_\tau^+. \tag{2.32}$$

Commutativity is a direct corollary of the following exchange relation,

---

[3]In the absence of a general proof, we regard uniqueness of factorization as a technical assumption.

[4]To underline the fact that $\chi_\tau$ acts on a Cartesian product of auxiliary spaces only, we denote it with a different symbol. In the case of $\mathcal{X} \cong \mathcal{Y}$, $\chi_\tau$ becomes equivalent to $\psi_\tau$.

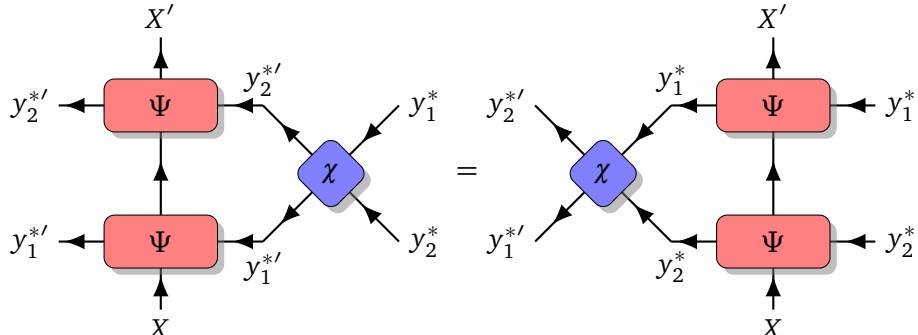

Figure 6: Repeated use of the Yang-Baxter relation (2.27) yields the composite exchange relation (2.33). The fixed points $y_{1,2}^*$ of the right-hand side are mapped by $\chi$ to the fixed points $y_{1,2}^{*\prime}$ of the left-hand side. Matching maps in the vertical space gives the commutation of two left row propagators (2.34). Time-step parameters are omitted for clarity.

$$\Psi_\tau^{\mathbf{32}} \circ \Psi_{\tau'}^{\mathbf{31}} \circ \chi_{\tau'-\tau}^{\mathbf{21}} = \chi_{\tau'-\tau}^{\mathbf{21}} \circ \Psi_{\tau'}^{\mathbf{31}} \circ \Psi_\tau^{\mathbf{32}}, \tag{2.33}$$

obtained by an $L$-fold application of the local Yang-Baxter relation (2.27), where the bold superscript **3** pertains to the global physical phase space $\mathcal{X}^L$, see Figure 6 for a diagrammatic representation. Given a pair of fixed points $(y_2^*, y_1^*)$, evaluating the left-hand side of Eq. (2.33) for an arbitrary $X$, the intertwiner $\chi_\tau$ outputs the pair of fixed points $(y_1^{*\prime}, y_2^{*\prime})$ and, vice-versa, evaluating the left-hand side of Eq. (2.33) with $(y_1^{*\prime}, y_2^{*\prime})$ as an input, we obtain $(y_2^*, y_1^*)$. Notice that the resulting map from $\mathcal{X}^L$ to itself (in the vertical direction, see Figure 6) is precisely the left chiral propagator given by Eq. (2.11). Therefore, Eq. (2.33) implies[5]

$$\Phi_\tau^- \circ \Phi_{\tau'}^- = \Phi_{\tau'}^- \circ \Phi_\tau^-, \tag{2.34}$$

and, by repeating the same reasoning, the same conclusion holds for $\Phi_\tau^+$. Commutation between the left and right chiral propagators is an immediate consequence of the inversion relation (2.12).

**Canonicity**  A map is canonical whenever it preserves the canonical Poisson brackets

$$\{q_i, p_j\} = \delta_{i,j}, \qquad \{q_i, q_j\} = \{p_i, p_j\} = 0. \tag{2.35}$$

Denoting the vector of canonical coordinates of a point $x \in \mathcal{X}$ by $\mathbf{x} = (q_1, p_1, \ldots q_{n_\mathcal{X}}, p_{n_\mathcal{X}})$ and similarly by $\mathbf{y}$ the coordinates for $y \in \mathcal{Y}$, a map $\psi$ preserves the Poisson bracket (2.35) precisely when the associated Jacobian matrix

$$\nabla_\psi = \begin{bmatrix} \frac{\partial \mathbf{x}'}{\partial \mathbf{x}} & \frac{\partial \mathbf{x}'}{\partial \mathbf{y}} \\ \frac{\partial \mathbf{y}'}{\partial \mathbf{x}} & \frac{\partial \mathbf{y}'}{\partial \mathbf{y}} \end{bmatrix}, \tag{2.36}$$

satisfies the condition

$$\nabla_\psi^T \, \Omega \, \nabla_\psi = \Omega, \tag{2.37}$$

where $\Omega = \mathbb{1} \otimes i\sigma^2$ (with the Pauli $y$-matrix $\sigma^2$) is the symplectic unit of dimension $n_\mathcal{X}$. Our construction involves canonical time propagators, i.e. maps $\Phi_\tau^\pm$ satisfying

$$\nabla_{\Phi_\tau^\pm}^T \, \Omega \, \nabla_{\Phi_\tau^\pm} = \Omega. \tag{2.38}$$

Since establishing Eq. (2.38) for $\Phi_\tau^\pm$ in full generality appears rather difficult, we demonstrate canonicity of $\Phi_\tau^\pm$ case by case, see Section 4.

---

[5]Assuming that the fixed points defining the propagators are not mixed by the intertwiner $\chi$.

**Parity**    Provided that there exists a parametrization of the spectral parameter $\lambda$ such that the physical and auxiliary Lax matrices satisfy an inversion identity of the form

$$\mathcal{L}_\lambda \mathcal{L}_{-\lambda} \simeq \mathcal{L}^a_\lambda \mathcal{L}^a_{-\lambda} \simeq \mathbb{1}, \tag{2.39}$$

we find the two-body map obeys the following inversion formula

$$\psi_{-\tau} \circ \psi_\tau = \mathrm{id}, \tag{2.40}$$

as a direct corollary of applying Eq. (2.39) to the exchange relation (2.2). The inversion relation (2.40) allows us to resolve the iteration (2.14) for the right propagator $\Phi^+_\tau$ using $\psi_{-\tau}$ directly instead of its inverse $\psi^{-1}_\tau$. The two chiral propagators $\Phi^\pm_\tau$ are then related by the space inversion transformation $\mathcal{P}$,

$$\mathcal{P}(X) = (x_L, \dots, x_2, x_1), \tag{2.41}$$

which is clearly involutive, $\mathcal{P} \circ \mathcal{P} = \mathrm{id}$. By applying (2.41) to Eqs. (2.7) or (2.13), and comparing the resulting local maps (2.8) and (2.14), we deduce that

$$\mathcal{P} \circ \Phi^\pm_{\tau, \Xi} \circ \mathcal{P} = \Phi^\mp_{-\tau, -\mathcal{P}(\Xi)}. \tag{2.42}$$

Notice that the inhomogeneities transform as well, as indicated by the second subscript.

# 3    Integrable fishnet circuits

Equipped with a pair of integrable chiral propagators $\Phi^\pm_\tau$, we now proceed to define a discrete-time evolution on the physical lattice $\mathcal{X}^L$. Due to the discrete space-time structure resembling a fishnet, see Figure 3, we dub the resulting systems "fishnet circuits".

Having established the elementary one-step chiral propagators $\Phi^\pm_\tau$ are in mutual involution for any value of $\tau$, the most general integrable fishnet circuit is simply obtained by a sequential iteration of $\Phi^\pm_\tau$, where we are free to choose an arbitrary sequence of time-step parameters. Motivated by physical applications, we however consider only a restricted family of fishnet circuits obeying additional symmetry and regularity properties. We are particularly interested in constructing integrable space-time discretizations of integrable Hamiltonian flows, which dictates the choice of fixed points to insure a smooth continuous-time limit.

**Combined one-step propagator**    Seeking to remedy the inherent chirality of the propagators $\Phi^\pm_\tau$, we combine them into a single propagator $\Phi_\tau : \mathcal{X}^L \to \mathcal{X}^L$ of the form

$$\Phi_\tau = \Phi^-_{-\tau} \circ \Phi^+_\tau. \tag{3.1}$$

Being the composition of canonical maps, the constructed propagator $\Phi_\tau$ is canonical itself. Owing to Eq. (2.32), $\Phi_\tau$ forms a one-parameter commuting family of maps,

$$\Phi_\tau \circ \Phi_{\tau'} = \Phi_{\tau'} \circ \Phi_\tau. \tag{3.2}$$

Moreover, from Eq. (2.12) it follows that

$$\Phi_\tau \circ \Phi_{-\tau} = \mathrm{id}. \tag{3.3}$$

The definition (3.1) is motivated by noting that, provided the inversion relation (2.40) is satisfied, the combined propagator $\Phi_\tau$ transforms under parity (2.42) as

$$\mathcal{P} \circ \Phi_{\tau, \Xi} \circ \mathcal{P} = \Phi_{\tau, -\mathcal{P}(\Xi)}, \tag{3.4}$$

The combined propagator $\Phi_\tau$ then becomes achiral, i.e. symmetric under parity $\mathcal{P}$, when $\Xi + \mathcal{P}(\Xi) = 0$, which is trivially satisfied in the homogeneous case $\Xi = 0$.

**Finite-time propagator** Given an arbitrary sequence of time-steps $\boldsymbol{\tau} = (\tau_1, \tau_2, \ldots, \tau_t)$, we furthermore introduce the finite-time propagator $\Phi_{\boldsymbol{\tau}}^{(t)} : \mathcal{X}^L \to \mathcal{X}^L$ with $t \in \mathbb{Z}_+$, given by an $n$-fold composition of one-step propagators

$$\Phi_{\boldsymbol{\tau}}^{(t)} = \Phi_{\tau_t} \circ \ldots \circ \Phi_{\tau_2} \circ \Phi_{\tau_1}, \tag{3.5}$$

Owing to Eq. (3.2), individual components can be freely permuted without altering $\Phi_{\boldsymbol{\tau}}^{(t)}$. Using the inversion relation (3.3), the definition (3.5) can be readily extended to negative values of $t$ upon inverting the time-step parameters, $\Phi_{\boldsymbol{\tau}}^{(-t)} = \Phi_{-\boldsymbol{\tau}}^{(t)}$, so that $\Phi_{\boldsymbol{\tau}}^{(t)} \circ \Phi_{\boldsymbol{\tau}}^{(-t)} = \mathrm{id}$. We denote the time-evolution of an observable $\mathcal{O} : \mathcal{X}^L \to \mathbb{C}$ as

$$\mathcal{O}(t) = \mathcal{O} \circ \Phi_{\boldsymbol{\tau}}^{(t)}, \tag{3.6}$$

where we have suppressed the dependence of the time-evolved observable on $\boldsymbol{\tau}$. The charges $Q_k^{(n)}$ introduced in Eq. (2.23) are preserved by $\Phi_{\boldsymbol{\tau}}^{(t)}$,

$$Q_k^{(n)} \circ \Phi_{\boldsymbol{\tau}}^{(t)} = Q_k^{(n)}. \tag{3.7}$$

**Ultralocal conservation laws** Certain integrable models admit additional local conservation laws not generated by the transfer functions. These are commonly related to the manifest global symmetries of a system and may be interpreted as Noether charges. To this end, we assume that the map $\psi$ (2.3) admits one or more ultralocal conservation laws of the form

$$q_0(x') - q_0(x) + q_0(y') - q_0(y) = 0, \tag{3.8}$$

with density $q_0 : \mathcal{X} \to \mathbb{C}$. Note that Eq. (3.8) takes the form of a local continuity equation, with charge densities of auxiliary variables corresponding precisely to local densities of physical Noether currents (whose sign depends on chirality, i.e. direction of propagation). The total ultralocal charge $Q_0 : \mathcal{X}^L \to \mathbb{C}$ reads

$$Q_0(X) = \sum_{\ell=1}^{L} q_0(x_\ell). \tag{3.9}$$

By summing the local continuity equation (3.8) over the entire chain and using that chiral propagators are defined by imposing the fixed point condition, we readily deduce that $Q_0$ is also preserved by the time evolution with $\Phi_{\boldsymbol{\tau}}^{(t)}$,

$$Q_0 \circ \Phi_{\boldsymbol{\tau}}^{(t)} = Q_0. \tag{3.10}$$

**Invariant measure** Given an arbitrary countable basis of conservation laws $\{H_k\}_{k=0}^{\infty}$ we define a family of stationary phase-space measures on $\mathcal{X}^L$ of the form

$$\mathrm{d}\Gamma_{\boldsymbol{\mu}} = Z_{\boldsymbol{\mu}}^{-1} e^{\boldsymbol{\mu} \cdot H} \prod_{\ell=1}^{L} \mathrm{d}\rho_\ell, \tag{3.11}$$

depending on chemical potentials $\boldsymbol{\mu} = (\mu_0, \mu_1, \mu_2, \ldots)$ coupling to $H = (H_0, H_1, H_2, \ldots)$, with normalization $Z_{\boldsymbol{\mu}} = \int_{\mathcal{X}^L} e^{\boldsymbol{\mu} \cdot H} \prod_{\ell=1}^{L} \mathrm{d}\rho_\ell$ corresponding to the partition function and $\mathrm{d}\rho_\ell$ denoting the uniform measure on the $\ell$-th copy of $\mathcal{X}$. Canonicity of $\Phi_{\boldsymbol{\tau}}^{(t)}$ and conservation of $H$ implies the measure (3.11) is invariant under the time-evolution by $\Phi_{\boldsymbol{\tau}}^{(t)}$. The average of an observable $\mathcal{O} : \mathcal{X}^L \to \mathbb{C}$ with respect to the invariant measure (3.11) reads

$$\langle \mathcal{O} \rangle_{\boldsymbol{\mu}} = \int_{\mathcal{X}^L} \mathcal{O} \, \mathrm{d}\Gamma_{\boldsymbol{\mu}}. \tag{3.12}$$

# 4 Examples

To exemplify the general construction presented in Section 2 and Section 3, we now exhibit two particular examples. Our first example is the widely studied Toda chain [32,33], a prototypical model of nonlinearly coupled oscillators, which permits a fully analytic construction, including specification of the fixed points (as in eg. [34]). The second important example is the anisotropic Landau-Lifshitz model on a lattice, where we first prove the existence of fixed points, construct them analytically and also show how to efficiently approximate them using fixed point iteration.

## 4.1 Toda lattice

The Toda lattice describes a one-dimensional chain of particles interacting via a nearest neighbor exponential potential, governed by the Hamiltonian

$$H_{\text{Toda}} = \sum_{\ell=1}^{L} \tfrac{1}{2} p_\ell^2 + e^{q_{\ell+1} - q_\ell}, \tag{4.1}$$

subjected to the periodic boundary conditions $\ell \equiv L + \ell$. The displacements and momenta $(q_\ell, p_\ell) \in \mathbb{R}^2$ obey the canonical Poisson brackets

$$\{q_\ell, p_{\ell'}\} = \delta_{\ell,\ell'}, \quad \{q_\ell, q_{\ell'}\} = \{p_\ell, p_{\ell'}\} = 0. \tag{4.2}$$

For convenience, we introduce the exponential of the displacement, $\mathrm{x}_\ell = e^{q_\ell} \in \mathbb{R}^+$, in terms of which the Poisson brackets reads

$$\{\mathrm{x}_\ell, p_{\ell'}\} = \mathrm{x}_\ell \delta_{\ell,\ell'}, \quad \{\mathrm{x}_\ell, \mathrm{x}_{\ell'}\} = \{p_\ell, p_{\ell'}\} = 0 \tag{4.3}$$

while the continuous-time evolution is given by

$$\frac{\mathrm{d}}{\mathrm{d}t}\mathrm{x}_\ell = \mathrm{x}_\ell p_\ell, \qquad \frac{\mathrm{d}}{\mathrm{d}t}p_\ell = \mathrm{x}_{\ell+1}/\mathrm{x}_\ell - \mathrm{x}_\ell/\mathrm{x}_{\ell-1}. \tag{4.4}$$

**Lax matrix**   The Lax matrix of the Toda lattice reads [35]

$$\mathcal{L}_\lambda(\mathrm{x}, p) = \begin{bmatrix} \lambda + p & -\mathrm{x} \\ 1/\mathrm{x} & 0 \end{bmatrix}. \tag{4.5}$$

There are various compatible auxiliary Lax matrices $\mathcal{L}^a$ to choose from. In order to be able to retrieve the Hamiltonian limit, we here pick [35,36]

$$\mathcal{L}_\lambda^a(\mathrm{y}, k) = \begin{bmatrix} \lambda + \mathrm{y}k & -\mathrm{y} \\ k & -1 \end{bmatrix}, \tag{4.6}$$

depending on variables $\mathrm{y}$ and $k$ that satisfy the same Poisson brackets as $\mathrm{x}$ and $p$, see Eq. (4.3), whereas $\mathrm{x}, p$ Poisson commute with $\mathrm{y}, k$. If the latter were considered as physical degrees of freedom, the Lax matrix (4.6) generates the integrable Hamiltonian hierarchy corresponding to the discrete self-trapping (DST) model. Both Lax matrices $\mathcal{L}$ and $\mathcal{L}^a$ are compatible and satisfy the Sklyanin bracket (2.1) with the classical $r$-matrix[6]

$$r_\lambda = \frac{\Pi}{\lambda}, \tag{4.7}$$

where $\Pi(u \otimes v) = v \otimes u$ represents the permutation in $\mathbb{C}^2 \otimes \mathbb{C}^2$ which, expressed in terms of Pauli matrices $\{\sigma^a\}_{a=0}^3$, reads explicitly $\Pi = \tfrac{1}{2}\Sigma_{a=0}^3 \sigma^a \otimes \sigma^a$.

---

[6]The Toda Lax matrix is obtained from the DST Lax matrix by contracting the oscillator algebra into the Euclidean Lie–Poisson algebra [36].

**Two-body map**     The exchange relation (2.2) is solved by $(x', p', y', k') = \psi_\tau(x, p, y, k)$ where

$$x' = y, \tag{4.8}$$
$$p' = x/y + yk - \tau, \tag{4.9}$$
$$y' = x(p + \tau - x/y), \tag{4.10}$$
$$k' = 1/x. \tag{4.11}$$

The inverse map $(x', p', y', k') = \psi_\tau^{-1}(x, p, y, k)$ is given by

$$x' = 1/k, \tag{4.12}$$
$$p' = yk + 1/kx - \tau, \tag{4.13}$$
$$y' = x, \tag{4.14}$$
$$k' = x^{-1}(p + \tau - 1/kx). \tag{4.15}$$

A straightforward calculation shows that the map $\psi_\tau$ is not canonical. Nevertheless, the chiral propagators $\Phi_\tau^\pm$ are canonical, as already established by Sklyanin [23, 24, 35].

**Fixed points**     We now seek to determine the fixed points of the elementary one-step chiral propagators constructed from the two-body map $\psi_\tau$ given by Eqs. (4.8)-(4.11). We begin by identifying the fixed points of the left chiral propagator $\Phi_\tau^-$. Firstly, it is immediately evident from Eqs. (4.11) that the fixed-point value of the auxiliary momentum $k$ is simply given by $k^* = 1/x_L$. To compute the fixed-point value of the conjugate variable y, we cast Eq. (4.10) as a matrix equation

$$\alpha_{\ell-1} v_\ell = \mathcal{F}_\ell v_{\ell-1}, \tag{4.16}$$

where $v_\ell = [\alpha_\ell, 1]^T$ with $\alpha_\ell = y_\ell/x_\ell$ and

$$\mathcal{F}_\ell = \begin{bmatrix} p_\ell + \tau & -x_{\ell+1}/x_\ell \\ 1 & 0 \end{bmatrix}, \tag{4.17}$$

while identifying $x_\ell \equiv x_{L+\ell}$. Iterating Eq. (4.16) along the entire chain amounts to multiplying the initial vector $v_0$ by a unimodular matrix $\mathcal{F}^- = \mathcal{F}_L \ldots \mathcal{F}_2 \mathcal{F}_1$, namely

$$A v_L = \mathcal{F}^- v_0, \tag{4.18}$$

where $A = \prod_{\ell=1}^{L} \alpha_{\ell-1}$. Imposing the periodicity condition $v_L = v_0$ is then equivalent to finding a right eigenvector $v = [y^*/x_L, 1]^T$ of $\mathcal{F}^-$,

$$\mathcal{F}^- v = Av. \tag{4.19}$$

We therefore have two fixed points $y_\pm^*$, one for each eigenvector $v_\pm$ with eigenvalues $A_\pm$. Note that if we wish to preserve the model's phase space, the fixed point value $y^*$ must be positive real. However, the eigenvector $v$ is not guaranteed to be real and a suitable fixed point does not exist for all initial conditions. When real eigenvectors exists they define two distinct chiral propagators, depending on which fixed point we use. Since $\mathcal{F}^-$ is unimodular, i.e. $A_+ A_- = 1$, we assume that $|A_+| \geq |A_-|$ so that $|A_+| > 1$ while $|A_-| < 1$. Both fixed points can be numerically approximated using fixed-point iteration, as discussed in more detail in Section 4.2. In particular, the eigenvector with eigenvalue $A_+$ corresponds to an attractive fixed point while the eigenvector with eigenvalue $A_-$ corresponds to a repulsive fixed point.

    For completeness, we also consider the fixed points of the right chiral propagator $\Phi_\tau^+$. We first note that the fixed-point value of the auxiliary displacement y is simply given by $y^* = x_1$.

To determine the fixed-point value of the auxiliary momentum $k$, we note that Eq. (4.15) can be cast as a matrix equation

$$\beta_\ell u_{\ell-1} = \mathcal{F}_\ell u_\ell, \tag{4.20}$$

where $u_\ell = [\beta_\ell, 1]^T$ with $\beta_\ell = k_\ell x_{\ell+1}$, c.f. Eq. (4.20). Introducing similarly $B = \prod_{\ell=1}^L \beta_\ell$, an iteration with the unimodular matrix $\mathcal{F}^+ = \mathcal{F}_1 \mathcal{F}_2 \ldots \mathcal{F}_L$ of an initial vector $u_L$ can be cast as

$$B u_0 = \mathcal{F}^+ u_L. \tag{4.21}$$

The periodicity condition $u_L = u_0$ is then equivalent to finding a right eigenvector $u = [k_L x_1, 1]^T$ of $\mathcal{F}^+$, c.f. Eq.(4.19)

$$\mathcal{F}^+ u = B u. \tag{4.22}$$

**Continuous-time limit**    We now demonstrate how the Toda chain can be retrieved by taking the $\tau \to \infty$ limit. To this end, we consider the left propagator $\Phi_\tau^-$ with $\Xi = 0$ and rewrite Eqs. (4.9) and (4.10) as

$$x'_{\ell+1}/x_\ell = p_\ell + \tau - \frac{x_\ell/x_{\ell-1}}{x'_\ell/x_{\ell-1}}, \tag{4.23}$$

$$p'_\ell = x_\ell/x'_\ell + x'_\ell/x_{\ell-1} - \tau, \tag{4.24}$$

which we subsequently expand for large $\tau$, obtaining

$$x'_{\ell+1}/x_\ell = p_\ell + \tau - \tau^{-1} x_\ell/x_{\ell-1} + \mathcal{O}(\tau^{-2}), \tag{4.25}$$

$$p'_\ell = x_\ell/x'_\ell + \tau^{-1}(x_\ell/x_{\ell-1} - x_{\ell-1}/x_{\ell-2}). \tag{4.26}$$

Next, we consider the right propagator $\Phi_\tau^+$ with $\Xi = 0$ and rewrite Eqs. (4.15) and (4.13) as

$$x_\ell/x'_{\ell-1} = p_{\ell-1} + \tau - \frac{x_{\ell+1}/x_\ell}{x_{\ell+1}/x'_\ell}, \tag{4.27}$$

$$p'_\ell = x'_\ell/x_\ell + x_{\ell+1}/x'_\ell - \tau, \tag{4.28}$$

whose expansion at large $\tau$ reads

$$x_\ell/x'_{\ell-1} = p_{\ell+1} + \tau - \tau^{-1} x_{\ell+1}/x_\ell + \mathcal{O}(\tau^{-2}), \tag{4.29}$$

$$p'_\ell = x_\ell/x'_\ell + \tau^{-1}(x_{\ell+1}/x_\ell - x_{\ell+2}/x_{\ell+1}). \tag{4.30}$$

Denoting the $\Phi_\tau$-propagated variable by a double prime, the composed time-propagator $\Phi_\tau$ to the leading order in $1/\tau$ reads

$$x''_\ell/x_\ell = -1 + 2\tau^{-1} p_\ell + \mathcal{O}(\tau^{-2}), \tag{4.31}$$

$$p''_\ell = p_\ell - 2\tau^{-1}(x_{\ell+1}/x_\ell - x_\ell/x_{\ell-1}) + \mathcal{O}(\tau^{-2}). \tag{4.32}$$

Apart from the $-1$ term on the right of Eq. (4.31), Eqs. (4.31) and (4.32) match with the continuous-time evolution (4.4). This small mismatch is remedied by considering instead the composition of two composed propagators with equal step-steps, that is $\Phi_\tau \circ \Phi_\tau$. Denoting the corresponding propagated variables by hatted symbols $\hat{\bullet}$, we thus find

$$(\hat{x}_\ell - x_\ell)/\delta = x_\ell p_\ell + \mathcal{O}(\delta^2), \tag{4.33}$$

$$(\hat{p}_\ell - p_\ell)/\delta = x_{\ell+1}/x_\ell - x_\ell/x_{\ell-1} + \mathcal{O}(\delta^2), \tag{4.34}$$

where $\delta = -4\tau^{-1}$. Clearly now in the continuous-time limit $\delta \to 0$ the doubled composed propagator recovers the Hamiltonian dynamics given by Eq. (4.4).

## 4.2   Anisotropic lattice Landau-Lifshitz model

In the anisotropic lattice Landau-Lifshitz model [3,37,38], local spin degrees of freedom live on the unit two-sphere $\mathbf{S} = (S^1, S^2, S^3) \in \mathcal{S}^2$, and satisfy the $SO(3)$ Lie-Poisson bracket,

$$\{S^a_\ell, S^b_{\ell'}\} = \varepsilon^{abc} S^c_\ell \delta_{\ell,\ell'}, \tag{4.35}$$

where $\varepsilon^{abc}$ is the Levi-Civita symbol.

**Lax operator**   The auxiliary Lax operator of the anisotropic Landau-Lifshitz model coincides with the Lax operator i.e. $\mathcal{L}^a = \mathcal{L}$, satisfies the inversion relation (2.39) and is given in Ref. [3]

$$\mathcal{L}_\lambda(\mathbf{S}) = \frac{1}{\sinh(2\mathrm{i}\varrho\lambda)} \begin{bmatrix} \sinh(2\mathrm{i}\varrho\lambda + \varrho S^3) & F_\varrho(S^3)S^- \\ F_\varrho(S^3)S^+ & \sinh(2\mathrm{i}\varrho\lambda - \varrho S^3) \end{bmatrix}, \tag{4.36}$$

where $S^\pm = S^1 \pm \mathrm{i}S^2$, $F_\varrho(s) = \sqrt{(\sinh^2\varrho - \sinh^2[\varrho s])/(1-s^2)}$. The parameter $\varrho$ pertains to axial anisotropy. We distinguish three 'regimes' of the model: (i) the *easy-axis* regime for $\varrho \in \mathbb{R}^+$, (ii) the *easy-plane* regime for $\varrho = \mathrm{i}\gamma \in \mathrm{i}(-\pi/2, \pi/2)$ and (iii) the *isotropic point* for $\varrho \to 0$. The Lax matrix (4.36) satisfies the inversion relation (2.39) and Sklyanin's quadratic bracket (2.1) with the trigonometric classical $r$-matrix of the form

$$r_\lambda = \frac{\varrho}{2\sin(2\varrho\lambda)} \left( \sigma^1 \otimes \sigma^1 + \sigma^2 \otimes \sigma^2 + \cos(\varrho\lambda)\sigma^3 \otimes \sigma^3 \right). \tag{4.37}$$

The Lax matrix enjoys two involutive symmetries,

$$\overline{\mathcal{L}_\lambda} = \sigma^2 \mathcal{L}_{\overline{\lambda}} \sigma^2, \qquad \mathcal{L}_\lambda^T = \sigma^2 \mathcal{L}_{-\lambda} \sigma^2, \tag{4.38}$$

where $\overline{\bullet}$ indicates complex conjugation, which are inherited by the monodromy matrix,

$$\overline{\mathcal{M}_\lambda} = \sigma^2 \mathcal{M}_{\overline{\lambda}} \sigma^2, \qquad \mathcal{M}_\lambda^T = \sigma^2 \mathcal{M}_{-\lambda} \sigma^2 \circ \mathcal{P}. \tag{4.39}$$

**Two-body map**   The solution to the local exchange relation (2.2) with the Lax matrix (4.36) is already known from Ref. [13], where it is also shown that the resulting Yang–Baxter map $\psi_\tau$ is canonical. For completeness, we here collect the main expressions.
It is convenient to introduce an anisotropic stereographic variable $\zeta \in \mathbb{C}$

$$\zeta = \sqrt{\frac{\sinh[\varrho(1-S^3)]}{\sinh[\varrho(1+S^3)]}} \frac{S^-}{\sqrt{1-(S^3)^2}}, \tag{4.40}$$

with the inverse

$$S^3 = \frac{1}{2\varrho} \log\left( \frac{|\zeta|e^{-\varrho} + |\zeta|^{-1}e^{\varrho}}{|\zeta|e^{\varrho} + |\zeta|^{-1}e^{-\varrho}} \right), \qquad S^\pm = \frac{\mathrm{Re}(\zeta) \mp \mathrm{i}\mathrm{Im}(\zeta)}{|\zeta|} \sqrt{1-(S^3)^2}. \tag{4.41}$$

The anisotropic variable $\zeta$ reduces to the standard stereographic variable $z = \frac{S^-}{1+S^3} \in \mathbb{C}$ in the isotropic limit $\varrho \to 0$. In terms of the variable $\zeta$ the Lax operator (4.36) takes the form

$$\mathcal{L}_\lambda(\zeta) = \frac{\begin{bmatrix} \sinh[\varrho(1+2\mathrm{i}\lambda)] - \sinh[\varrho(1-2\mathrm{i}\lambda)]\zeta\overline{\zeta} & \sinh(2\rho)\zeta \\ \sinh(2\rho)\overline{\zeta} & \sinh[\varrho(1+2\mathrm{i}\lambda)]\zeta\overline{\zeta} - \sinh[\varrho(1-2\mathrm{i}\lambda)] \end{bmatrix}}{\sinh(2\mathrm{i}\varrho\lambda)\sqrt{(e^{2\rho} + \zeta\overline{\zeta})(e^{-2\rho} + \zeta\overline{\zeta})}}, \tag{4.42}$$

while the two-body map $(\zeta'_1, \zeta'_2) = \psi_\tau(\zeta_1, \zeta_2)$ takes the form (derived from the one given in Ref. [39] after elementary manipulations)

$$\zeta'_1 = \mathcal{H}_{\tau,\varrho}(\zeta_1, \zeta_2), \qquad \zeta'_2 = \mathcal{H}_{\tau,\varrho}(\zeta_2, \zeta_1), \tag{4.43}$$

where

$$\mathcal{H}_{\tau,\varrho}(\zeta_1,\zeta_2) = \frac{\zeta_2 \sinh(2\varrho) + \zeta_1 |\zeta_2|^2 \sinh[2\varrho(1-i\tau)] - \zeta_1 \sinh(2i\tau\varrho)}{\zeta_1 \overline{\zeta}_2 \sinh(2\varrho) + \sinh[2\varrho(1-i\tau)] - |\zeta_2|^2 \sinh(2i\tau\varrho)}. \tag{4.44}$$

At the isotropic point, i.e. in the limit $\varrho \to 0$, the map simplifies considerably

$$\mathcal{H}_{\tau,0}(z_1,z_2) = \frac{z_2 + z_1 |z_2|^2 (1-i\tau) - i\tau z_1}{z_1 \overline{z}_2 + (1-i\tau) - i\tau |z_2|^2} \tag{4.45}$$

and can be written compactly in terms of canonical spin variables [11, 40]

$$\psi_\tau(\mathbf{S}_1, \mathbf{S}_2) = \frac{1}{\tau^2 + \Delta^2} \left( \Delta^2 \mathbf{S}_2 + \tau^2 \mathbf{S}_1 + \tau \mathbf{S}_2 \times \mathbf{S}_1, \Delta^2 \mathbf{S}_1 + \tau^2 \mathbf{S}_2 + \tau \mathbf{S}_1 \times \mathbf{S}_2 \right), \tag{4.46}$$

with $\Delta^2 = \frac{1}{2}(1 + \mathbf{S}_1 \cdot \mathbf{S}_2)$. The map $\psi_\tau$ is canonical for all values of $\varrho$ and $\tau$, and its canonicity immediately carries over to $\Psi_\tau$ defined in Eq. (2.8),

$$\nabla_\Psi^T \Omega \nabla_\Psi = \Omega, \tag{4.47}$$

since $\Psi_\tau$ is obtained by compositions of $\psi_\tau$. Remarkably, a direct calculation reported in Appendix A shows that the elementary chiral propagators $\Phi_\tau^\pm$ remain canonical after projecting $\Psi_\tau$ and its inverse to the physical phase space, cf. Eqs. (2.11) and (2.18).

**Existence of fixed points and the Lefschetz-Hopf theorem**   The construction of Section 2 crucially relies on the existence of a fixed point $y_\tau^*$ of $\pi_\mathcal{Y} \circ \Psi_\tau$. Before proceeding with an analytical construction of the fixed points of the anisotropic Landau-Lifshitz model, we demonstrate their existence on general topological grounds which can be generalized to other models.

By invoking the Lefschetz-Hopf theorem [41] we can prove the existence of fixed points. The theorem relates the number of fixed points of a continuous map $f : \mathcal{Y} \to \mathcal{Y}$ from a compact topological space onto itself to the so-called Lefschetz number $\Lambda_f \in \mathbb{Z}_{\geq 0}$,

$$\sum_{y \in \text{fix}(f)} \text{ind}(f, y) = \Lambda_f, \tag{4.48}$$

where $\text{fix}(f) = \{y \in \mathcal{Y} | f(y) = y\}$ is the (assumed to be finite) set of the fixed points of $f$, and $\text{ind}(f, y) \in \mathbb{N}$ is the index associated to a given fixed point. Crucially, the Lefschetz number can be alternatively computed as the alternating sum of traces of maps induced by $f$ on the singular homology groups $H_k$ of $\mathcal{Y}$,

$$\Lambda_f = \sum_{k=0}^\infty (-1)^k \text{tr}[H_k(f, \mathcal{Y})]. \tag{4.49}$$

Presently, the local physical space is the unit two-sphere $\mathcal{Y} = \mathcal{S}^2$ whose only non-zero homology groups are $H_0$ and $H_2$. The function $\pi_{\mathcal{S}^2} \circ \Psi_\tau$ is invertible for $\tau \neq 0$. The trace of the induced map on $H_0$ is trivial

$$\text{tr}\left[H_0(\pi_{\mathcal{S}^2} \circ \Psi_\tau, \mathcal{S}^2)\right] = 1, \tag{4.50}$$

while the trace of the induced map on $H_2$ depends on whether the map is orientation-preserving or not. Noting that $\lim_{\tau \to \infty} \pi_{\mathcal{S}^2} \circ \Psi_\tau = \text{id}$ for arbitrary $X$, the map is continuously connected to the identity and therefore orientation preserving, yielding

$$\text{tr}\left[H_2(\pi_{\mathcal{S}^2} \circ \Psi_\tau, \mathcal{S}^2)\right] = 1. \tag{4.51}$$

The Lefschetz number follows from Eq. (4.49) and equals

$$\Lambda_{\pi_{\mathcal{S}^2} \circ \Psi_\tau} = 2. \tag{4.52}$$

This means that for all $X$ and $\tau$ there exists either a single fixed point with index two or a pair of fixed points with index one by the Lefschtz-Hopf theorem (4.48). Note that the theorem is not constructive, that is, it tells us nothing about how to find the fixed points.

**Fixed points of isotropic propagator** The fixed points of the isotropic Landau-Lifschitz model can be obtained explicitly due to the simple form of the isotropic Lax operator,

$$\mathcal{L}_\lambda^{\text{iso}}(\mathbf{S}) = \mathbb{1} + \frac{\mathbf{S} \cdot \boldsymbol{\sigma}}{2i\lambda}, \tag{4.53}$$

where $\boldsymbol{\sigma} = (\sigma^1, \sigma^2, \sigma^3)^T$ is a vector of Pauli matrices. Parametrizing the associated monodromy matrix $\mathcal{M}_{\lambda,\Xi}^{\text{iso}}(X) = \mathcal{L}_{\lambda-\xi_L}^{\text{iso}}(x_L) \dots \mathcal{L}_{\lambda-\xi_1}^{\text{iso}}(x_1)$ in the form

$$\mathcal{M}_\lambda^{\text{iso}} = \begin{bmatrix} a(\lambda) & b(\lambda) \\ c(\lambda) & d(\lambda) \end{bmatrix}, \tag{4.54}$$

the first involution (4.39) yields the following relations

$$\overline{a(\lambda)} = d(\overline{\lambda}), \qquad \overline{b(\lambda)} = -c(\overline{\lambda}). \tag{4.55}$$

We note that the monodromy matrix is a polynomial of degree $L$ in $1/\lambda$. On the other hand, considering the defining relation of the left propagator $\Phi_\tau^-$ (2.7) we have the relation,

$$\mathcal{M}_{\lambda^+}^{\text{iso}}(X') = \left[\mathcal{L}_{\lambda^-}^{\text{iso}}(\mathbf{S}_\tau^*)\right]^{-1} \mathcal{M}_{\lambda^+}^{\text{iso}}(X) \mathcal{L}_{\lambda^-}^{\text{iso}}(\mathbf{S}_\tau^*), \tag{4.56}$$

which would result in apparent poles of the monodromy matrix. To determine the poles we consider the group element $g \in SU(2)$ that aligns the fixed-point spin $\mathbf{S}_\tau^*$ with the third axis,

$$g(\mathbf{S}_\tau^* \cdot \boldsymbol{\sigma})g^{-1} = \sigma^3. \tag{4.57}$$

Introducing the rotated monodromy $\tilde{\mathcal{M}}_\lambda^{\text{iso}} = g\mathcal{M}_\lambda^{\text{iso}}g^{-1}$, Eq. (4.56) yields the relation

$$\tilde{\mathcal{M}}_{\lambda^+}^{\text{iso}}(X') = \left(\mathbb{1} + \sigma^3/2i\lambda^-\right)^{-1} \tilde{\mathcal{M}}_{\lambda^+}^{\text{iso}}(X)\left(\mathbb{1} + \sigma^3/2i\lambda^-\right), \tag{4.58}$$

where the 12 and 21 matrix elements of the right-hand side of Eq. (4.58) have poles located at $1/2\lambda_- = \mp i$ respectively

$$\left[\tilde{\mathcal{M}}_{\lambda^+}^{\text{iso}}(X)\right]_{12} = \tilde{b}(\lambda_+)\frac{i - 1/2\lambda_-}{i + 1/2\lambda_-}, \tag{4.59}$$

$$\left[\tilde{\mathcal{M}}_{\lambda^+}^{\text{iso}}(X)\right]_{21} = \tilde{c}(\lambda_+)\frac{i + 1/2\lambda_-}{i - 1/2\lambda_-}. \tag{4.60}$$

Absence of poles on the left-hand side of Eq. (4.58) requires that the off-diagonal elements of the rotated monodromy vanish at their respective poles, namely

$$\tilde{b}(\tau + i/2) = \tilde{c}(\tau - i/2) = 0. \tag{4.61}$$

To solve them we fix the $U(1)$-gauge freedom of $g$ and write the gauge transformation $g$ in terms of the stereographic projection $z \in \mathbb{C}$,

$$g(z) = \frac{1}{\sqrt{1 + z\bar{z}}}\begin{bmatrix} 1 & z \\ -\bar{z} & 1 \end{bmatrix}. \tag{4.62}$$

A direct calculations using now shows that the first condition (4.61) translate into the same quadratic equation for $z$ of the form

$$z^2 c(\tau + i/2) + z(a(\tau + i/2) - d(\tau + i/2)) - b(\tau + i/2) = 0, \tag{4.63}$$

which can be shown to be also equivalent to the second condition using Eq. (4.55). The quadratic equation (4.63) gives two solutions $z^-_{j\in\{1,2\}}$, with corresponding fixed-point spins $\mathbf{S}^-_{j,\tau} \equiv \mathbf{S}^*_\tau(z^-_j)$ that follow from Eq. (4.57) with

$$\mathbf{S}(z) \cdot \boldsymbol{\sigma} = \frac{1}{1 + z\bar{z}} \begin{bmatrix} 1 - z\bar{z} & 2z \\ 2\bar{z} & z\bar{z} - 1 \end{bmatrix}. \tag{4.64}$$

Repeating the analysis for the fixed points of the right propagator $\Phi^+_\tau$ we find the conditions

$$\tilde{b}(\tau - i/2) = \tilde{c}(\tau + i/2) = 0, \tag{4.65}$$

both of which are equivalent to the quadratic equation

$$z^2 c(\tau - i/2) + z(a(\tau - i/2) - d(\tau - i/2)) - b(\tau - i/2) = 0, \tag{4.66}$$

for $z^+_j$ that characterize the fixed-point spins $\mathbf{S}^+_{j,\tau} \equiv \mathbf{S}^*_\tau(z^+_j)$. Using the relations (4.55) it is easy to show that the quadratic Eq. (4.63) for $\Phi^-_\tau$ and Eq. (4.66) for $\Phi^+_\tau$ are related by the variable transformation $z \to -1/\bar{z}$, which implies that the solutions satisfy

$$z^-_{1,2} \bar{z}^+_{2,1} = -1. \tag{4.67}$$

The justification for pairing different solution indices in Eq. (4.67) is given in the next paragraph when we consider the fixed points of the anisotropic propagator.

**Fixed points of the anisotropic propagator**   A complementary approach that generalizes to deriving the fixed points of the anisotropic propagator is closely related to the construction of fixed points of the Toda model described in Section 4.1.

Starting with the left-to-right propagation defined in Eq. (2.7), we note that the iteration of the anisotropic map $\psi_\tau$ (4.44) in anisotropic stereographic coordinates $\zeta$ can be written as

$$\alpha_\ell u_{\ell+1} = \mathcal{L}_{\tau+i/2}(\zeta_\ell) u_\ell, \tag{4.68}$$

where $u_\ell = [\zeta^a_\ell, -1]^T$ with $\zeta^a_\ell$ the anisotropic stereographic variable of the $\ell$-th auxiliary spin while $\alpha_\ell \in \mathbb{C}$ is constant whose precise from is irrelevant and

$$\mathcal{L}_{\tau+i/2}(\zeta) = \frac{\begin{bmatrix} \sinh(2i\tau\varrho) - \sinh[2\varrho(1 - i\tau)]\zeta\bar{\zeta} & \sinh(2\rho)\zeta \\ \sinh(2\rho)\bar{\zeta} & \sinh(2i\tau\varrho)\zeta\bar{\zeta} - \sinh[2\varrho(1 - i\tau)] \end{bmatrix}}{\sinh[2i\varrho(\tau + i/2)]\sqrt{(e^{2\rho} + \zeta\bar{\zeta})(e^{-2\rho} + \zeta\bar{\zeta})}}. \tag{4.69}$$

The map $\pi_{\mathcal{Y}} \circ \Psi_\tau$ is realized by passing an initial vector $u_0$ across the monodromy matrix

$$Au_L = \mathcal{M}_{\tau+i/2}(X) u_0, \tag{4.70}$$

where $A = \prod_{\ell=1}^L \alpha_{\ell-1}$. Imposing the periodicity condition $u_L = u_0$ then amounts to finding a right eigenvector $u = [\zeta^-, -1]^T$ of the monodromy matrix

$$Au = \mathcal{M}_{\tau+i/2}(X) u. \tag{4.71}$$

The two eigenvectors $u_{j\in\{1,2\}}$ with eigenvalues $A_j$ of the monodromy matrix correspond to two fixed points that define the left propagators $\Phi^-_\tau$. Since there are no additional reality conditions on the deformed stereographic variables, the two fixed points exists for all physical spins $X$ in

agreement with the Lefschetz-Hopf theorem, unlike in the Toda model, where a reality condition precluded the existence of fixed points for some inputs. We note that the relation (4.70) can be interpreted as a power method for estimating the eigenvalues and eigenvectors, as we further elaborate when discussing fixed point iteration. For generic $X$, the eigenvalues' moduli are non-degenerate and we order them as $|A_1| > |A_2|$. The eigenvectors $u_{1,2}$ then correspond to the attractive and repulsive fixed points of the power method iteration respectively.

We similarly analyze the left-to-right propagator defined in Eq. (2.13) by noting that Eq. (4.68) can also encode iteration of $\psi_{-\tau}$ in the opposite direction by

$$\beta_\ell u_{\ell-1} = \mathcal{L}_{-\tau+\mathrm{i}/2}(\zeta_\ell)u_\ell, \tag{4.72}$$

where again $u_\ell = [\zeta_\ell^a, -1]^T$. Recalling the inversion relation $\mathcal{L}_{-\lambda}\mathcal{L}_\lambda \simeq \mathbb{1}$ and using the first involutive symmetry (4.39), the left-to-right iteration of $\pi_\mathcal{Y} \circ \Psi_\tau^{-1}$ is realized by

$$Bv_0 = \left[\mathcal{M}_{\tau+\mathrm{i}/2}(X)\right]^{-1} v_L, \tag{4.73}$$

where $v_\ell = [1, \overline{\zeta}_\ell^+]^T$. Multiplying by the monodromy matrix and imposing the periodicity condition $v_0 = v_L$ we find the eigenvalue relation

$$B^{-1}v = \mathcal{M}_{\tau+\mathrm{i}/2}(X)v, \tag{4.74}$$

which coincides with the eigenvalue relation (4.71) upon identifying

$$\zeta_{1,2}^- \overline{\zeta}_{2,1}^+ = -1, \tag{4.75}$$

cf. Eq. (4.67), where the identification of repulsive and attractive fixed points comes from noting that $B_j = A_j^{-1}$. From Eq. (4.40) we then readily conclude that the attractive fixed point of the left propagator correspond to the spatial inversion around the origin of the repulsive fixed point of the right propagator and vice-versa

$$\mathbf{S}_{1,2;\tau}^\pm = -\mathbf{S}_{2,1;\tau}^\mp. \tag{4.76}$$

To make a connection with the twisted monodromy matrix approach we note that using the parametrization of the monodromy matrix (4.54) in the eigenvalue Eqs. (4.71) and (4.74) we find precisely the quadratic Eqs. (4.63) and (4.66) in $\zeta^\mp$ respectively after eliminating the eigenvalues which now holds for all anisotropies $\varrho$.

**Fixed-point iteration**   Instead of determining the fixed points analytically, it is sometimes more convenient to approximate them using a numerical algorithm. As our construction outlined in Section 2 already suggests, a fixed point of $\pi_\mathcal{Y} \circ \Psi_\tau$ can be numerically determined by means of fixed-point iteration [42], see also Ref. [43] where it is shown that fixed-point iteration in the box-ball model converges in a single iteration.

Starting from a given an initial condition $y_0^{(0)}$ and fixed physical input variables $X$, we iterate the function $\pi_\mathcal{Y} \circ \Psi_\tau$ as shown in Figure 7

$$y_0^{(n+1)} = \pi_\mathcal{Y} \circ \Psi_\tau(X, y_0^{(n)}), \tag{4.77}$$

The iteration of Eq. (4.77) converges towards a fixed point $y_\tau^*$ (2.10)

$$y_\tau^*(X) = \lim_{n\to\infty} y_0^{(n)}. \tag{4.78}$$

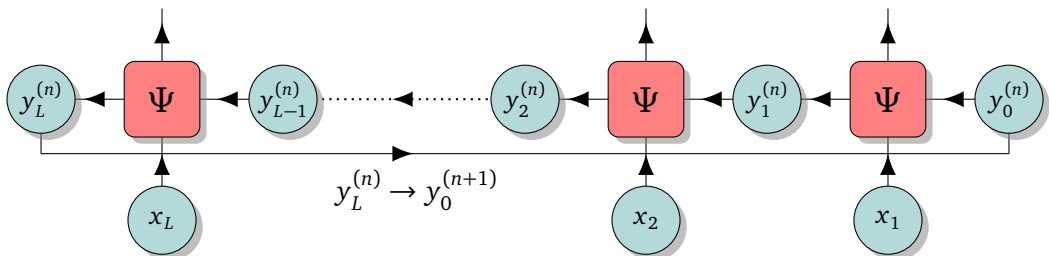

Figure 7: The function $\pi_\mathcal{Y} \circ \Psi_\tau$ with fixed physical variables $X$ is iterated according to Eq. (4.77) by repeatedly imposing the periodic closure condition in the horizontal direction by setting $y_0^{(n+1)} = y_L^{(n)}$.

In general one might expect the iteration to converge to different fixed points depending on the initial condition $y_0^{(0)}$. However, as noted in the analysis of the fixed points of the anisotropic propagator, a generic initial condition always converges to the attractive fixed point, irrespectively of the values of anisotropy $\varrho$ and the time-step $\tau$. Similarly, performing the iteration with the inverse of $\pi_\mathcal{Y} \circ \Psi_\tau$ instead of $\Psi_\tau$, yields the other, repulsive fixed point.

    While the convergence properties of the anisotropic Landau-Lifschitz model's fixed points follow from their explicit construction, it is worthwhile to briefly discuss the convergence properties of fixed-point iteration in general. A map $f : \mathcal{Y} \to \mathcal{Y}$ is contracting if it satisfies

$$\|f(y_1) - f(y_2)\|_\mathcal{Y} \leq s\|y_1 - y_2\|_\mathcal{Y}, \tag{4.79}$$

for some $s \in [0, 1)$ where $\| \bullet \|_\mathcal{Y} : \mathcal{Y} \to \mathbb{R}_0^+$ specifies a norm in $\mathcal{Y}$. By the Banach fixed-point theorem [44], iteration of a continuous contracting map yields a unique fixed point for every initial condition. If the two-body map $\pi_\mathcal{Y} \circ \psi_\tau : \mathcal{Y} \to \mathcal{Y}$ with fixed $x$ was contracting for all $x \in \mathcal{X}$, then the composite map $\pi_\mathcal{Y} \circ \Psi_\tau$ with fixed $X$ would also be trivially contracting, implying both the existence and uniqueness of the fixed point. However, the two-body map (4.43) with the first (i.e. vertical) spin fixed is contracting only in the neighborhood of the attractive fixed point and not on the entire sphere.

**Convergence of fixed point iteration at the isotropic point**     For simplicity, we now focus on the isotropic map (4.46), and also give a direct demonstration that the outlined fixed-point iteration (4.77) with physical spins $X$ sampled from a maximum entropy measure always converges towards the attractive fixed point in the limit of long spin chains, by showing that the map is contracting on average.

Suppose we start the iteration from two generic initial spins $y_0$ and $y_0'$. Using Eq. (4.46), a direct calculation shows that

$$\|y_{\ell+1} - y_{\ell+1}'\|^2 = e^{s_\ell}\|y_\ell - y_\ell'\|^2, \tag{4.80}$$

where

$$e^{s_\ell} = \frac{\tau^2(1 + \tau^2)}{(\tau^2 + \Delta_\ell^2)(\tau^2 + \Delta_\ell'^2)}, \tag{4.81}$$

with $\Delta_\ell^2 = (1 + x_\ell \cdot y_{\ell-1})/2$ and $\|y\| = \sqrt{y \cdot y}$ is the Euclidean norm. Telescoping Eq. (4.80) across a chain of length $L$ and subsequently taking the logarithm we have

$$L^{-1} \log \frac{\|y_L - y_L'\|^2}{\|y_0 - y_0'\|^2} = \sum_{\ell=0}^{L-1} s_\ell. \tag{4.82}$$

Since $x_\ell$ are independently distributed, so are the factors $s_\ell$, and in the limit of a long chain $L \to \infty$ the sum on the right-hand side converges to the average value by the law of large numbers,

$$\lim_{L \to \infty} L^{-1} \log \frac{\|y_L - y_L'\|^2}{\|y_0 - y_0'\|^2} = \bar{s}, \tag{4.83}$$

where the average value is given explicitly by

$$\bar{s} = \frac{1}{4\pi} \int d\rho_{\mathcal{S}^2} \, s_\ell = 2 - (1 + 2\tau^2) \log(1 + 1/\tau^2). \tag{4.84}$$

Noting that $\log x \geq 2(x-1)/(x+1)$ for $x \geq 1$ we have $\bar{s} < 0$ which shows that the iteration (4.80) is on average contracting, demonstrating that any two initial vectors converge to the same point in the case of long isotropic chains.

**Continuous-time limit** Since the auxiliary Lax matrix $\mathcal{L}^a$ coincides with the physical Lax matrix $\mathcal{L}$, the two-body map evaluated at $\tau = 0$ becomes a permutation

$$\psi_0(x, y) = (y, x), \tag{4.85}$$

as implied by the local exchange relation (2.2). The corresponding homogeneous ($\Xi = 0$) left and right chiral propagators reduce to cyclic shifts

$$\Phi_0^\pm = \eta^\pm, \tag{4.86}$$

where $\eta^\pm(X) = (x_{1\pm1}, x_{2\pm1}, \ldots, x_{L\pm1})$ stand for the right and left cyclic shifts, respectively. The homogeneous chain is therefore symmetric under spatial reflection (i.e. parity $\mathcal{P}$), and close to the identity map for small $\tau$, with the expansion

$$\Phi_\tau = \mathrm{id} + \phi_\tau + \mathcal{O}(\tau^2). \tag{4.87}$$

Here $\phi_\tau = \tau\{\bullet, H_{\mathrm{LLL}}\}$ is the Hamiltonian flow generated by a logarithmic Hamiltonian

$$H_{\mathrm{LLL}} = -\sum_{\ell=1}^{L} \log h(\mathbf{S}_\ell, \mathbf{S}_{\ell+1}), \tag{4.88}$$

where we identify $\ell \equiv \ell + L$. The local two-body interaction results from a straightforward but tedious calculation, reading

$$h(\mathbf{S}_\ell, \mathbf{S}_{\ell+1}) = F_\varrho(S_\ell^3) F_\varrho(S_{\ell+1}^3) \left(S_\ell^1 S_{\ell+1}^1 + S_\ell^2 S_{\ell+1}^2\right) - \cosh\left(\frac{\varrho}{2}\left[S_\ell^3 + S_{\ell+1}^3\right]\right)$$
$$+ \frac{1}{2} \cosh\left(\frac{\varrho}{2}\left[S_\ell^3 + S_{\ell+1}^3 + 2\right]\right) + \frac{1}{2} \cosh\left(\frac{\varrho}{2}\left[S_\ell^3 + S_{\ell+1}^3 - 2\right]\right). \tag{4.89}$$

This shows that the constructed integrable fishnet circuit serves as a particular integrable space-time discretization of the corresponding continuous-time (i.e. Hamiltonian) lattice dynamics. It is instructive to compare it the known and more standard 'brickwork' integrable discretization [10–13], which also reduces to the same continuous-time dynamics in the limit of small time steps. Despite that, two constructions are inequivalent. The main distinguished property of the fishnet circuits is that elementary one-step propagators mutually commute for any arbitrary time-step parameters (2.32), ensuring conservation of the entire transfer function (2.19) of the continuous-time dynamics generated by the lattice Hamiltonian or any combination of the charges $Q_k$.[7]

---

[7] In the anisotropic Landau–Lifshitz model, the only nontrivial coefficient of the spectral curve is $\mathcal{T}^{(1)}(\lambda)$ and hence only a single tower of charges appears.

By contrast, the elementary Floquet propagators arising in the brickwork circuits conserve the staggered version of a staggered transfer function, yielding two independent towers of conservation laws that are not in involution with the charges of the Hamiltonian model. Put differently, the Floquet propagators do not commute at different values of the time-step parameter, i.e. different time-steps invariably break integrabilty of the time evolution. In the language of statistical mechanics, our construction can be view as a 'row-to-row transfer matrix', as opposed to the brickwork construction that corresponds to a 'diagonal-to-diagonal transfer matrix'.

Moreover, as we demonstrate in Section 5, the freedom to choose arbitrary inhomogeneities in the time direction allows us to define a stochastic integrable dynamics with peculiar dynamical properties which cannot be realized in the brickwork setting.

## 5   Stochastic integrable dynamics

This Section is devoted to the study of spin dynamics in the anisotropic lattice Landau-Lifshitz model introduced in Section 4.2. In particular, we consider space-homogeneous models (i.e. set $\Xi = 0$) and specialize to spin transport in maximum-entropy equilibrium states. The main object of our interest is the dynamical structure factor of magnetization density,

$$C_\tau(\ell, t) = \langle S_\ell^3(t) S_0^3(0) \rangle_{\mu=0}^c, \tag{5.1}$$

where $\langle XY \rangle^c = \langle XY \rangle - \langle X \rangle \langle Y \rangle$ is the connected two-point correlation function, while $\tau$ indicates explicit dependence of the time-propagation with $\Phi_\tau^{(t)}$ on the chosen sequence of time-steps $\tau$.

### 5.1   Stochastic propagator

Dynamical properties of the uniform fishnet circuits with equal parameters $\tau$ at every step are qualitatively similar to those found in the brickwork discretizations of the Hamiltonian model. Integrable fishnet circuit nevertheless allow for a richer phenomenology as (owing to commutativity of propagators $\Phi_\tau$, see Eq. (3.2)) we now have the freedom of picking arbitrary time-step parameter while still preserving integrability.

To exemplify this explicitly, we consider random fishnet circuits by drawing the time-steps from a distribution $\Omega : \mathbb{R}^t \to \mathbb{R}_{\geq 0}$. More specifically, we consider the dynamics under an ensemble-averaged propagator

$$\overline{\Phi}_\Omega^{(t)} = \int \mathrm{d}^t \tau \, \Omega(\tau) \Phi_\tau^{(t)}. \tag{5.2}$$

For simplicity, we specialize to ensembles with uncorrelated time-steps for which the $t$-step probability distribution $\Omega$ factorizes into a product of one-step distributions $\omega : \mathbb{R} \to \mathbb{R}_{\geq 0}$,

$$\Omega(\tau) = \prod_{j=1}^t \omega(\tau_j). \tag{5.3}$$

We denote the corresponding averaged propagator by $\overline{\Phi}_\omega^{(t)}$. Note that while commutativity of the combined propagators (3.2) is not essential for defining the ensemble-averaged propagator (5.2), it greatly facilitates the following computations.

**Example: Discrete two-point measure**    It is instructive to consider ensemble-averaging of a dynamics whose time-steps are sampled from a discrete two-point measure

$$\omega_2(s) = \frac{1-\eta}{2}\delta(s+\tau) + \frac{1+\eta}{2}\delta(s-\tau), \qquad -1 \le \eta \le 1, \tag{5.4}$$

where $\eta\tau = \int \mathrm{d}s\, s\,\omega_2(s)$ and $(1-\eta^2)\tau^2 = \int \mathrm{d}s\,(s-\eta\tau)^2\omega_2(s)$ are the time-step's mean and variance respectively. Using the commutation of combined propagators (3.1) and the inversion property (3.3), the ensemble-averaging in Eq. (5.2) simplifies to a straightforward combinatorial sum of combined propagators with a different number of time steps

$$\overline{\Phi}_{\omega_2}^{(t)} = 2^{-t}\sum_{t'=-t}^{t}\binom{t}{\frac{t-t'}{2}}(1-\eta)^{\frac{t-t'}{2}}(1+\eta)^{\frac{t+t'}{2}}\Phi_\tau^{(t')}, \tag{5.5}$$

where we use the convention that binomial coefficients with half-integer arguments vanish.

**Asymptotic propagator for $\eta \ne 0$**    The asymptotic behavior of the sum (5.5) can be studied using the De Moivre-Laplace theorem

$$\binom{n}{k}p^k q^{n-k} \simeq \mathcal{N}_k(np, npq) \tag{5.6}$$

for $n \to \infty$ with $p+q=1$ and $k-np \sim n^{1/2}$ where $\mathcal{N}_x(\mu,\sigma^2) = e^{-\frac{(x-\mu)^2}{2\sigma^2}}/\sqrt{2\pi\sigma^2}$ is a Gaussian distribution. Using the De Moivre-Laplace theorem (5.6) the sum (5.5) becomes

$$\overline{\Phi}_{\omega_2}^{(t)} \simeq \int_{-t}^{t}\mathrm{d}t'\mathcal{N}_{t'}(\eta t,(1-\eta^2)t)\,\Phi_\tau^{(\lfloor t'\rfloor)}. \tag{5.7}$$

The behavior of the integral (5.7) crucially depends on the value of $\eta$. Setting $t' = ut$, we note that the Gaussian factor tends to a $\delta$-function, $t^{-1}\mathcal{N}_u(\eta,(1-\eta^2)t^{-1}) \to \delta(u-\eta)$, resulting in

$$\overline{\Phi}_{\omega_2}^{(t)} \simeq \Phi_\tau^{(\lfloor t\eta\rfloor)}. \tag{5.8}$$

The ensemble-averaged time-extensive propagator at $t$ steps for $\eta \ne 0$ is then simply the time-extensive propagator at $\lfloor t\eta\rfloor$ steps. In other words, for $\eta \ne 0$ the leading order of the asymptotic dynamics is dominated by a deterministic dynamics on an $\eta$-rescaled time-scale, see the middle panel of Figure 4 for an example.

**Asymptotic propagator for $\eta = 0$**    The asymptotic behavior of the dynamics is distinct for $\eta = 0$ where Eq. (5.8) indicates that the propagator's leading order vanishes. The leading correction is obtained by setting $t' = \zeta t^{1/2}$, yielding

$$\overline{\Phi}_{\omega_2}^{(t)} \simeq \int_{\mathbb{R}}\mathrm{d}\zeta\,\mathcal{N}_\zeta(0,1)\,\Phi_\tau^{(\lfloor\zeta t^{1/2}\rfloor)}. \tag{5.9}$$

Some comments on Eq. (5.9) are in order. Unlike for $\eta \ne 0$, where the leading order of the ensemble-averages propagator was asymptotically deterministic, the leading order ensemble-averaged propagator for $\eta = 0$ remains stochastic, driven by fluctuations of the difference of positively- and negatively-signed time-steps. In a given realization of $t$ time steps, a typical fluctuations will have an excess of $\mathcal{O}(t^{1/2})$ of a time-steps with given sign and the resulting stochastic dynamics occurs on this timescale, see the right panel of Figure 4 for an example.

## 5.2 Brownian dynamics in the continuous-time limit

The example presented in Section 5.1 uses a simple discrete measure that permits a straightforward combinatorial analysis. Here we show that as a consequence of the central limit theorem similar behavior occurs in the $\tau \to 0$ limit for distributions $\omega$ characterized by a finite mean $\mu = \int s\omega(s)ds$ and variance $\sigma^2 = \int (s-\mu)^2 \omega(s)ds$.

The expansion of the combined propagator to linear order in the time-step reads

$$\Phi_\tau = \mathrm{id} + \phi_\tau + \mathcal{O}(\tau^2), \tag{5.10}$$

where, crucially, the time-steps of the linearized propagator are additive,

$$\phi_\tau + \phi_{\tau'} = \phi_{\tau+\tau'}. \tag{5.11}$$

Now consider the expansion of the extensive propagator (3.5) to linear order with all time-steps of the same order, $\tau_j \sim \mathcal{O}(\tau)$

$$\Phi_\tau^{(t)} = \mathrm{id} + \sum_{j=1}^t \phi_{\tau_j} + \mathcal{O}(\tau^2) = \mathrm{id} + \phi_{\sum_{j=1}^t \tau_j} + \mathcal{O}(\tau^2), \tag{5.12}$$

where we made use of the additivity property (5.11). To analyze the ensemble averaged continuous-time propagator we distinguish distributions with zero and non-zero means.

**Continuous-time propagator for $\mu \neq 0$**  In the scaling limit $\tau \to 0$, $t \to \infty$, with $T = \tau t$ kept constant, we infer from the expansion (5.12) that

$$\lim_{\substack{t\to\infty,\tau\to0 \\ t\tau=T}} \overline{\Phi}_\omega^{(t)} = \mathrm{id} + \phi_{\mu T}, \tag{5.13}$$

where we noted that the sum $\sum_{j=1}^t \tau_j$ with $\tau_j \sim \omega$ converges to $\mu T$ by the law of large numbers. The result (5.13) shows that in the continuous-time limit the ensemble-averaged dynamics obtained from a distribution with a non-zero mean produces a deterministic continuous-time dynamics, c.f. Eq. (5.8).

**Continuous-time propagator for $\mu = 0$**  For $\mu = 0$ the leading order in the continuous-time limit vanishes, see Eq. (5.13). To capture the leading-order correction we send $\tau \to 0$ and $t \to \infty$ such that $T = t\tau^2$ is constant. The expansion (5.12) now gives

$$\lim_{\substack{t\to\infty,\tau\to0 \\ t\tau^2=T}} \overline{\Phi}_\omega^{(t)} = \mathrm{id} + \int_{\mathbb{R}} d\zeta \, \mathcal{N}_\zeta(0,\sigma^2) \phi_{\zeta T^{1/2}}, \tag{5.14}$$

where we noted that the quantity $T^{-1/2}\sum_{j=1}^t \tau_j$ for $\tau_j \sim \omega$ becomes normally distributed (with variance $\sigma^2$) in the limit $t \to \infty$ with $T = t\tau^2$ held constant by the central limit theorem. The result (5.14) shows for distributions with zero mean the continuous-time dynamics becomes Brownian, cf. Eq. (5.9). Since integrability is preserved despite the stochastic rescaling of time, such dynamics support 'Brownian soltions', i.e. solitons undergoing Brownian motion, see the right panel of Figure 4.

**Dressing the dynamical structure factor** To exemplify the dynamical manifestations of the Brownian dynamics (5.14) we consider the ensemble-averaged dynamical structure factor

$$\overline{C}_{\Omega}(\ell, t) = \int d^t \tau \, \Omega(\tau) C_{\tau}(\ell, t). \tag{5.15}$$

We are primarily interested in its large space-time behavior $C(\ell, t) \simeq t^{-1/z} f(\ell/t^{-1/z})$, where $C(\ell, t)$ is the dynamical structure factor of deterministic continuous-time dynamics with $z$ and $f$ the corresponding dynamical exponent and scaling function respectively.

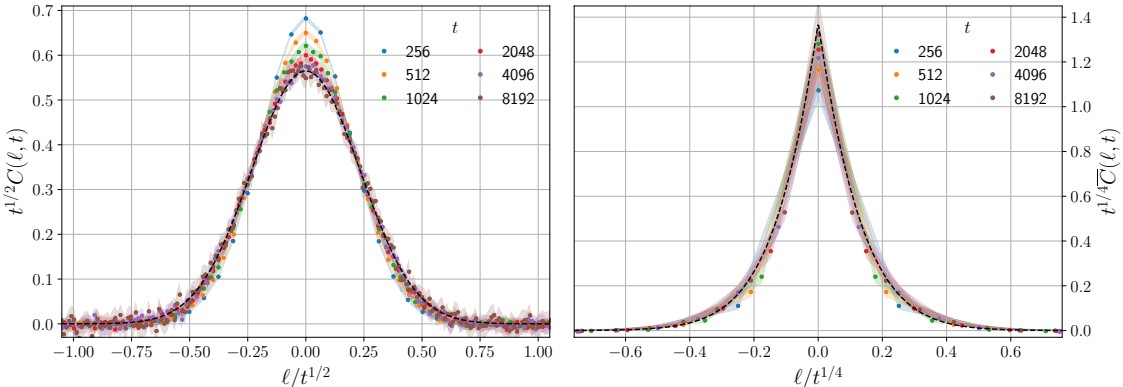

Figure 8: Quasi-continuous spin dynamics ($\tau = 0.01$) of the integrable fishnet circuit discretization of the Landau-Lifschitz model in the easy-axis regime with anisotropy $\varrho = 1$: (left panel) The deterministic dynamics is characterized by the algebraic dynamical exponent $z = 2$ and asymptotically Gaussian (dashed black line) spin structure factor (colored points); (right panel) the stochastic Brownian dynamics (colored points) exhibits the double dynamical exponent $\overline{z} = 4$, with the structure factor asymptotically approaching the M-Wright distribution (dashed black line) obtained by a stochastic dressing of a Gaussian structure factor using Eq. (5.17). Shaded regions show two standard deviations neighborhoods.

Using Eq. (5.14) we find the asymptotic behavior of Brownian continuous-time dynamics

$$\overline{C}(\ell, t) \simeq t^{-1/\overline{z}} \overline{f}(\ell t^{-1/\overline{z}}), \tag{5.16}$$

with a doubled dynamical exponent $\overline{z} = 2z$ and the scaling function

$$\overline{f}(s) = \int_{\mathbb{R}} d\zeta |\zeta|^{-1/z} \mathcal{N}_{\zeta}(0, \sigma^2) f(s/|\zeta|^{-1/z}). \tag{5.17}$$

An interesting example of the dressed dynamical structure factor (5.17) is the easy-axis regime of the anisotropic lattice Landau-Lifschitz model in maximum entropy states. As shown in the left panel of Figure 8, the spin structure factor of deterministic dynamics is asymptotically Gaussian with dynamical exponent $z = 2$. The corresponding Brownian dynamics have a dynamical structure factor $\overline{z} = 4$ with a dressed structure factor that follows from (5.17) and is given by the M-Wright function [45, 46]. The M-Wright functions has recently appeared as the distribution of typical charge fluctuations in charged single-file models [47–49] and has also been numerically observed to describe typical spin fluctuations in the easy-axis regime [50, 51]. While its appearance in Brownian dynamics of the same model is suggestive we cannot presently establish a general connection between the two results.

# 6   Conclusion

We have presented an algebraic construction of a family of integrable discrete space-time dynamical systems, starting from a discrete zero-curvature condition for a pair of Lax operators, iterations of which from left to right and vice-versa define the corresponding chiral propagators. Their periodic closure renders them isospectral, while the set-theoretic Yang-Baxter equations ensures their commutativity for arbitrary time-steps, a characteristic of Bäcklund transformations.

To account for the intrinsic chirality of the propagators we defined the combined propagator by composing a left and a right propagator with opposite time-steps. The time-extensive dynamics of the systems is obtained by stacking combined propagators with arbitrary time-steps, which allows for efficient numerical simulations of integrable systems with arbitrary time-varying coupling constants. Remarkably, the discrete-time dynamics conserves the charges of continuous-time Hamiltonian dynamics of the corresponding Lax operator.

By considering i.i.d. time-steps we obtained a class of integrable stochastic dynamical systems which combine the usually distinct phenomenology of both domains. By manipulating the time-step distribution, we demonstrated a transition from deterministic to stochastic soliton dynamics using a two-point distribution as an example. For distributions with a vanishing mean, the leading-order dynamics cease to be ballistic and instead occur on a sub-ballistic scale, as showcased by Brownian (i.e. diffusing) solitons in the easy-plane regime of the lattice Landau-Lifshitz model. As demonstrated, this behavior is universal in the stochastic continuous-time limit for all time-step distributions with finite variance.

Several points warrant further investigations. The existence of a fixed point was pivotal in the construction of the chiral propagators. For all examples considered in this work, their existence was shown by explicit construction and, for the anisotropic Landau-Lifschitz model, their existence demonstrated on general grounds using the Lefschetz-Hopf theorem which relates the number of fixed points of a map to the topological properties of the phase-space. Since the theorem is not constructive, we have also shown how to approximate fixed points numerically by fixed point iteration, which turned out to be remarkably efficient. It would be of interest to better understand the existence, number and convergence properties of fixed points for other spaces, for example symmetric spaces arising in matrix models with higher-rank groups symmetries [12].

The possibility of combining time-dependent and especially stochastic dynamics with intangibility calls for the extension of generalized hydrodynamics to this setting. Of particular interest is the case of integrable Brownian dynamics, for which we have microscopically derived the structure factor in terms of a dressing of the corresponding deterministic structure factor. A hydrodynamic approach might also shed some light on stochastic dynamics with generically distributed finite-size time-steps, where numerical simulations indicate differences compared to the two-point dressing considered here. While such dynamics are in principle amenable to a direct calculation, the mixing of dynamics with different time-steps makes microscopic progress difficult.

Brownian dynamics in the stochastic continuous-time limit arise due to universal Gaussian fluctuations described by the central limit theorem, which requires finiteness of the distribution's variance. It would be of interest to relax this assumptions and consider the resulting stochastic continuous-time dynamics, where we expect a one-parameter family of dynamics with distinct dynamical exponents generated by the corresponding Levy-stable distributions.

Lastly, while the present construction is purely classical, the involved algebraic structures have direct counterparts in the quantum setting, calling for a corresponding quantum construction which would facilitate the study of quantum Brownian solitons.

## Acknowledgments

ŽK thanks John Morgan and Juan Esteban Rodríguez Camargo for very useful discussions on the Lefschetz-Hopf theorem. We thank Gregoire Misguich for collaboration in the early stages of this work, Johannes Schmidt for collaboration on related topics and in particular the form (4.44) of the map.

**Funding information**  ŽK is supported by the Simons Foundation as a Junior Fellow of the Simons Society of Fellows (1141511). The work has been partly supported by European Research Council (ERC) through Advanced grant QUEST (Grant Agreement No. 101096208) (TP), as well as the Slovenian Research and Innovation agency (ARIS) through the Programs P1-0402 (EI and TP) and N1-0243 (EI).

## A  Canonicity of elementary propagators

We prove canonicity of the map $\Phi_\tau^-$ (2.11) in the anisotropic Landau-Lifschitz model by proving the condition (2.38). Canonicity of $\Phi_\tau^+$ follows analogously.

As noted, canonicity of the Landau-Lifschitz the two-body map (4.43) trivially extend to $\Psi_\tau$, see Eq. (4.47), whose Jacobian matrix has the block structure

$$\nabla_\Psi = \begin{bmatrix} \frac{\partial \mathbf{X}'}{\partial \mathbf{X}} & \frac{\partial \mathbf{X}'}{\partial \mathbf{y}} \\ \frac{\partial \mathbf{y}'}{\partial \mathbf{X}} & \frac{\partial \mathbf{y}'}{\partial \mathbf{y}} \end{bmatrix}, \tag{A.1}$$

where we denote the vector of canonical coordinates of a tuple of points as $\mathbf{X} = (\mathbf{x}_1, \mathbf{x}_2, \ldots, \mathbf{x}_L)$ and, to lighten notation, we also identify $y_0 \equiv y$ and $y_L \equiv y'$ and suppressed the $\tau$ parameter. To facilitate further computations we introduce the following index notation for components of $\mathbf{X}$ and $\mathbf{y}$

$$\mathbf{X} \to X_{i\alpha}, \qquad \mathbf{y} \to y_\alpha, \tag{A.2}$$

where latin indices indicate the many-body space position of $X$ and Greek indices run over the corresponding vector of canonical coordinates at the point $x_i$. We use the index summation convention of summing over repeated indices. For example, in index notation the Jacobian (A.1) reads

$$[\nabla_\Psi]_{i\alpha, j\beta} = \begin{bmatrix} \frac{\partial X'_{i\alpha}}{\partial X_{j\beta}} & \frac{\partial X'_{i\alpha}}{\partial y_\beta} \\ \frac{\partial y'_\alpha}{\partial X_{j\beta}} & \frac{\partial y'_\alpha}{\partial y_\beta} \end{bmatrix}. \tag{A.3}$$

A direct calculation now gives the block form of the condition for $\Psi$ (4.47)

$$\left[\nabla_\Psi^T \hat{\Omega} \nabla_\Psi\right]_{i\alpha, j\beta} = \begin{bmatrix} \frac{\partial X'_{k\gamma}}{\partial X_{i\alpha}}\frac{\partial X'_{k\eta}}{\partial X_{j\beta}} + \frac{\partial y'_\gamma}{\partial X_{i\alpha}}\frac{\partial y'_\eta}{\partial X_{j\beta}} & \frac{\partial X'_{k\gamma}}{\partial X_{i\alpha}}\frac{\partial X'_{k\eta}}{\partial y_\beta} + \frac{\partial y'_\gamma}{\partial X_{i\alpha}}\frac{\partial y'_\eta}{\partial y_\beta} \\ \frac{\partial X'_{k\gamma}}{\partial y_\alpha}\frac{\partial X'_{k\eta}}{\partial X_{j\beta}} + \frac{\partial y'_\gamma}{\partial y_\alpha}\frac{\partial y'_\eta}{\partial X_{j\beta}} & \frac{\partial X'_{k\gamma}}{\partial y_\alpha}\frac{\partial X'_{k\eta}}{\partial y_\beta} + \frac{\partial y'_\gamma}{\partial y_\alpha}\frac{\partial y'_\eta}{\partial y_\beta} \end{bmatrix} \Sigma_{\gamma\eta}, \tag{A.4}$$

$$[\hat{\Omega}]_{i\alpha, j\beta} = \begin{bmatrix} \delta_{ij}\Sigma_{\alpha\beta} & 0 \\ 0 & \Sigma_{\alpha\beta} \end{bmatrix}.$$

Identifying the respective blocks in Eq. (A.4) we find four identities which we will use later. Since $\Phi^-$ is defined by the fixed point $y^*$ we also compute its derivative from Eq. (2.10)

$$\frac{\partial y_\alpha^*}{\partial X_{i\beta}} = \frac{\partial y'_\alpha}{\partial X_{i\beta}} + \frac{\partial y'_\alpha}{\partial y_\gamma}\frac{\partial y_\gamma^*}{\partial X_{i\beta}} \tag{A.5}$$

and express

$$\frac{\partial y'_\alpha}{\partial X_{i\beta}} = \left(\delta_{\alpha\gamma} - \frac{\partial y'_\alpha}{\partial y_\gamma}\right)\frac{\partial y^*_\gamma}{\partial X_{i\beta}}. \tag{A.6}$$

Computing the Jacobian matrix of $\Phi^-$ defined in Eq. (2.11) yields

$$[\nabla_{\Phi^-}]_{i\alpha,j\beta} = \frac{\partial X'_{i\alpha}}{\partial X_{j\beta}} + \frac{\partial X'_{i\alpha}}{\partial y_\gamma}\frac{\partial y^*_\gamma}{\partial X_{j\beta}}. \tag{A.7}$$

The left-hand side of the condition (2.38) then reads

$$\left[\nabla^T_{\Phi^-}\hat{\Omega}\nabla_{\Phi^-}\right]_{i\alpha,j\beta} = \left(\frac{\partial X'_{k\gamma}}{\partial X_{i\alpha}}\frac{\partial X'_{k\eta}}{\partial X_{j\beta}} + \frac{\partial X'_{k\gamma}}{\partial y_\zeta}\frac{\partial y^*_\zeta}{\partial X_{i\alpha}}\frac{\partial X'_{k\eta}}{\partial X_{j\beta}}\right. \tag{A.8}$$
$$\left.+ \frac{\partial X'_{k\gamma}}{\partial X_{i\alpha}}\frac{\partial X'_{k\eta}}{\partial y_\zeta}\frac{\partial y^*_\zeta}{\partial X_{j\beta}} + \frac{\partial X'_{k\gamma}}{\partial y_\zeta}\frac{y^*_\zeta}{\partial X_{i\alpha}}\frac{\partial X'_{k\eta}}{y_\mu}\frac{\partial y^*_\mu}{\partial X_{j\beta}}\right)\Sigma_{\gamma\eta}.$$

Using the identities from the four blocks of Eq. (A.4) this becomes

$$\left[\nabla^T_{\Phi^-}\hat{\Omega}\nabla_{\Phi^-}\right]_{i\alpha,j\beta} = \delta_{ij}\Sigma_{\alpha\beta} - \left(\frac{\partial y'_\gamma}{\partial X_{i\alpha}}\frac{\partial y'_\eta}{\partial X_{j\beta}} + \frac{\partial y'_\gamma}{\partial y_\zeta}\frac{\partial y^*_\zeta}{\partial X_{i\alpha}}\frac{\partial y'_\eta}{\partial X_{j\beta}} + \frac{\partial y'_\gamma}{\partial X_{i\alpha}}\frac{\partial y'_\eta}{\partial y_\zeta}\frac{\partial y^*_\zeta}{\partial X_{j\beta}}\right. \tag{A.9}$$
$$\left.+ \frac{\partial y^*_\zeta}{\partial X_{i\alpha}}\frac{\partial y^*_\mu}{\partial X_{j\beta}}\frac{\partial y'_\gamma}{\partial y_\zeta}\frac{\partial y'_\eta}{\partial y_\mu} - \frac{\partial y^*_\gamma}{\partial X_{i\alpha}}\frac{\partial y^*_\eta}{\partial X_{j\beta}}\right)\Sigma_{\gamma\eta}.$$

Eliminating all instances of $\partial y'/\partial X$ via Eq. (A.6) and simplifying we find that all but the first term on the right-hand side of Eq. (A.9) cancel

$$\left[\nabla^T_{\Phi^-}\Omega\nabla_{\Phi^-}\right]_{i\alpha,j\beta} = \delta_{ij}\Sigma_{\alpha\beta}, \tag{A.10}$$

which proves the condition (2.38) for $\Phi^-$.

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
