# Peer review of "Integrable fishnet circuits and Brownian solitons"

_SciPost Physics, doi:SciPost Phys. 19, 027 (2025)_

## Round 1 · Referee Report · Yuan Miao (Referee 1) · 2025-1-29

Strengths

1-- The results are novel and important.

2-- The results can potentially lead to further investigations in the future.

3-- Both analytic and numeric results are solid.

Report

The authors studied a new type of discrete space-time evolution of classical integrable models, dubbed "integrable fishnet circuits". The integrability of the circuits is guaranteed by the Yang-Baxter map, and the authors demonstrated the construction in the examples of Toda lattice and anisotropic Landau-Lifshitz magnet. Due to the choice of the "time parameter" $\tau$, classical solitons that obey Brownian motion are discovered.

The results of the article are new and important, and it will lead to further investigations in various directions, such as the quantum counterparts and generalizations to other models related to higher-rank Lie algebras, etc. I believe that the article deserves to be published in SciPost Physics, once the comments below are properly addressed.

1-- The title "integrable fishnet circuits" needs to be clarified. In the field of integrable models, there exists already a terminology "Fishing-net diagram" (later popularized as "fishnet diagram"), a model of Feynman diagrams of fishnet type that can be solved by an integrable vertex model [R1]. Recently, more studies such as [R2] and [R3] have been performed in this field, which has a close relation to AdS/CFT integrability. Even though the content of the current article is not directly related to the research mentioned above, it is worth pointing out the existing literatures on those investigations and the difference/connection to current article, in order to avoid confusions to people who might be familiar with both topics.

2-- As mentioned in Sec. 1.1, the construction of fishnet circuits resembles the quantum transfer matrix formalism, can I understand the circuits as a "Trotterization" of the temporal direction, instead of the spatial direction of the brick-wall case?

3-- In the introduction, the authors cited the quantum brickwork circuits [10] and [14-17]. Of course the authors knew very well, I find it useful to mention the poineer work in the 1980s [R4, R5] as a reference. I also think that the authors could cite one of my papers [R6], where a generalization of the brickwork construction (generalized to arbitrary periodicity) is considered. This comment is only suggestive.

4-- In section 2, when constructing Lax matrices, we obtained a (n x n) Lax matrix, where n is the dimension of the auxiliary space. I wonder if the dimension of the auxiliary space n is fixed by any parameter, such as the phase space dimension $n_X$ or $n_Y$.

5-- I wonder if the zero-curvature condition (2.2) is the usual zero-curvature condition rotated by $\pi/4$.

6-- The line above (2.10), $\Psi_{\tau} : X^L \times Y \to X^L \times Y$ seems to be contradictory with (2.8) and (2.9). For instance, Eq. (2.9) implies that $\Psi_{\tau} : Y \to Y$, and the right-hand side of (2.8) implies that $\Psi_{\tau} : X^{L-1} \times Y \to X^{L-1} \times Y$.

7-- I'm confused about the number of functionally independent charges in footnote 2. My understanding is that the total dimension of the phase space of $X^L$ is $L \cdot n_X$.

8-- In the set-theoretical Yang-Baxter equation part, the YBE written down is of difference form. I'm wondering if it is possible to generalize to the case that is not of difference form. (It seems to me that it should be possible.)

9-- On page 13 above Eq. (3.11), if the number of sites $L$ is finite, do we still need to take into account infinitely-many charges ${H_k}_{k=0}^{\infty}$? Can I understand Eq. (3.11) as a "classical generalized Gibbs ensemble"?

10-- It is not clear what the acronym "DST Lax matrix" stands for in footnote 6. It would be better if the authors use the full name or explain it.

11-- I'm not sure if Eqs. (4.16) and (4.17) do not contain typos. If I'm not mistaken, Eq. (4.16) should read $\alpha_{\ell-1} \nu_{\ell} = \mathcal{F}_{\ell-1} \nu_{\ell-1}$. Eq. (4.17) should read $\mathcal{F}_{\ell} = \begin{pmatrix} p_{\ell}+ \tau & - x_{\ell} / x_{\ell+1} \\ 1 & 0 \end{pmatrix} $. I urge the authors to double-check the formulae mentioned.

12-- I do not understand the statement below Eq. (4.19), namely why fixed point value $y^\ast$ must be positive and real. It would be great if the authors could explain.

13-- The continuous-time limit on page 16: Naïvely, I expect the continuous-time limit to be $\tau \to 0$, while the number of time steps goes to infinity. Would it arrive at the same result as the limit on page 16.

14-- On page 18 about the existence of fixed points, I do not understand the meaning of the index of fix points. Especially in the line under Eq. (4.52), the authors mentioned the possibility of having a fixed point with index two. What exactly is the index, and what is the physical meaning of having a fixed point of index two? The Lefschetz-Hopf theorem is purely topological. I'm wondering if the existence of fixed points is only determined by the topology of the target space of the classical sigma model.

15-- I'm wondering if the authors could comment on the physical meaning of the fixed points in terms of the many-body dynamics (any counterpart in the quantum case? any relation to the MPO fixed point?), and in the case of multiple fixed points, do we expect different physics when choosing different fixed points?

16-- It seems to me that if we choose a different choice of the random time-steps (e.g. non-Gaussian), we could obtain different "Brownian motions" than the Gaussian ones. I'm wondering if that is the correct understanding.

[R1] A.B. Zamolodchikov, "Fishing-net" diagrams as a completely integrable system, Phys. Lett. B, 97, 63-66 (1980).

[R2] Ö. Gürdoğan and V. Kazakov, New Integrable 4D Quantum Field Theories from Strongly Deformed Planar $\mathcal{N}=4$ Supersymmetric Yang-Mills Theory, Phys. Rev. Lett. 117, 201602 (2016).

[R3] D. Chicherin, V. Kazakov, F. Loebbert, D. Müller and D. Zhong, Yangian Symmetry for Fishnet Feynman Graphs, Phys. Rev. D 96, 121901 (2017).

[R4] C. Destri and H.J. De Vega, Light-cone lattice approach to fermionic theories in 2D: The massive Thirring model, Nucl. Phys. B 290, 363-391 (1987).

[R5] C. Destri and H.J. De Vega, Integrable quantum field theories and conformal field theories from lattice models in the light-cone approach, Phys. Lett. B 201 (2), 261-268 (1988).

[R6] Y. Miao, V. Gritsev and D. Kurlov, The Floquet Baxterisation, SciPost Phys. 16, 078 (2024).

Requested changes

Please see the report above.

Recommendation

Ask for minor revision

  • validity: top
  • significance: high
  • originality: top
  • clarity: high
  • formatting: excellent
  • grammar: perfect

Author:  Žiga Krajnik  on 2025-03-12  [id 5285]

(in reply to Report 1 by Yuan Miao on 2025-01-29)

We thank the referee for their detailed reading of the manuscript and many useful comments and suggestions which have improved the manuscript. Their comments are addressed below:

  1. Since there already exist another class of integrable circuits, widely known as brickwork (or brickwall) circuits, we have similarly sought a short and suggestive name for the proposed circuit architecture. We have decided to go with 'fishnet' circuits, compatibly with their diagrammatic representation. While we are aware of 'fishnet Feynmann diagrams' arising in supersymmetric gauge theories, we cannot recognize any immediate connections beyond the shared graph structure. Curiously, there likewise exist `brickwall' type diagrams. Given that the scope of our work bears no overlap with the studies of integrability in the domain of quantum field theories, we do not see much benefits of expanding on our choice of nomenclature.
  2. Contrary to the brickwork circuits which directly implement the Trotter-Suzuki decomposition of the evolution operator, the fishnet circuits cannot (at least directly) be interpreted in that way. Despite that, we show that fishnet circuits can be understood as an integrable time-space discretization of Hamiltonian (i.e. continuous-time) dynamics which, in sharp contrast to integrable brickwork circuits, exactly preserve the hierarchy of Hamiltonians for an arbitrary discrete time-step. To use the language of vertex models, the brickwork geometry corresponds to diagonal-to-diagonal transfer matrices whereas our fishnet construction employs row-to-row transfer matrices.
  3. We thank the referee for pointing out these references which have now been incorporated into the text.
  4. We do not think there is any general connection. For instance, it is known that Lax operators with different dimensions (of the auxiliary matrix space) can encode the same dynamics.
  5. Yes, the zero-curvature condition used in our construction is precisely the one used in construction of brickwork circuits rotated by $\pi/4$ (see Figure 1), apart from an extra permutation of arguments.
  6. Eqs. 2.8 and 2.9 involve a slight abuse of notation - they iteratively define a family of maps (distinguished by the number of their arguments) by passing the auxiliary variable through the chain, see Figure 2. The map $\Psi_\tau$ in Eq. 2.10 is obtained after $L$ iterations (i.e. after passing the auxiliary variable across the entire length of the chain). We afforded this slight imprecision to avoid using unnecessarily heavy notation. To avoid confusion we have now added an additional subscript to distinguish these maps in 2.8 and 2.9, and similarly also for 2.14 and 2.15.
  7. We thank the referee for catching this typo which has been corrected.
  8. Yes, we believe a similar construction can be accomplished using a non-difference form of the Yang-Baxter relation provided the analogous non-difference zero-curvature condition can be resolved explicitly. That said, some of the outlined properties in the current construction may no longer hold.
  9. Yes, Eq. 3.11 can be indeed be viewed a classical generalized Gibbs ensemble. Notice however that the question of completeness of classical generalized Gibbs ensembles is not a trivial one, and it presently remains unclear clear how to identify and construct a complete set of charges that characterize local maximum-entropy ensembles.
  10. The acronym DST stands for `discrete self-trapping'. Its single occurrence has now been eliminated.
  11. We thank the referee for catching the mistake in 4.17 which has now been corrected. We have also made several other minor modifications in the paragraph regarding the fixed points of the Toda lattice to improve readability.
  12. Note that Eq. 4.8 gives $x^\prime_{\ell} = y_{\ell-1}$. Specializing to $\ell=1$ immediately gives that the fixed point $y^*$ must itself lie in the the phase space of the Toda lattice.
  13. One needs to take $\tau \to \infty$ to obtain the continuous-time limit, taking $\tau \to 0$ does not recover the correct limit. Note that this is one particularily of the Toda lattice; to obtain the local conserved quantities one likewise has to expand the transfer matrix around $\lambda \to \infty$.
  14. The index associated to a fixed point of a function counts the degeneracy of the fixed point under a generic perturbation of that function, see e.g. Chapter 3 of Differential Topology by V. Guillemin and A. Pollack. In the updated version we have corrected a mistake in the discussion of the number of fixed points. In particular, the index of a fixed point can be negative so that the Lefschtz-Hopf theorem only gives a lower bound on the number of fixed points. More concretely, the index of a fixed point is related to the topological behavior of the function in the vicinity of the fixed point. In two dimensions non-degenerate attractive and repulsive fixed points have index $1$ while non-degenerate saddle fixed points have index $-1$. However, since these considerations are not very essential to the manuscript (in fact, we also obtain the fixed points analytically), we now only note that the existence of a fixed point is guaranteed without discussing their multiplicity. In addition to topological properties of the phase-space, the existence of fixed points also depends on the homology properties of the local two-body map. In the present case of the 2-sphere, the relevant property is orientation preservation (or lack thereof) which determines the trace of the map induced on the second homology group. As noted, the map we consider is orientation-preserving whence the existence of a fixed point follows by the Lefschetz-Hopf theorem. On the other hand, an orientation-reversing map will produce a vanishing Lefschetz number and the existence of a fixed point is not guaranteed, as is easily seen by considering sphere inversion, i.e. the map $x \mapsto -x$ on the 2-sphere.
  15. The quantum analogue of the fixed-point condition would be to trace out the auxiliary (i.e. horizontal) space. However, it is presently not evident how the classical fixed-point condition arises in the semi-classical limit of the corresponding integrable quantum model (as the auxiliary dimension grows large). While different fixed points define distinct dynamics we have not encountered any meaningful physical differences, at least not in the considered cases. This is perhaps not surprising since the resulting dynamics always preserved the transfer matrix of the model. On the other hand, we cannot exclude that models with a richer phase-space and fixed-point structure might exhibit different behavior. We believe these questions merits further attention.
  16. The referee's intuition is correct. Our results were obtained assuming a finite variance of the time-step distribution. Relaxing this assumption and varying the tail exponent of the time-step distribution results in continuous-time dynamics with a smoothly varying dynamical exponent generated by Levy stable distributions by the generalized central limit theorem.

---

## Round 1 · Referee Report · Vincent Caudrelier (Referee 2) · 2025-1-30

Strengths

Please see my report for more details but overall strengths are:

  • New exciting results taking known methods and notions in a new direction
  • Great potential to bring together various communities and initiate further research.

Weaknesses

Again, more detail in my report but overall (minor) weaknesses are:

  • A limited rendition of how the work fits in the context of discrete integrable systems. This is not a judgement and not a problem (of course everyone knows certain areas better than others) and I have tried to make suggestions to improve the impact this work can have in the community of discrete integrable systems, at least.

  • Sometimes notations are a bit confusing. Again, I've tried to point out where this happens.

Report

Yes, this article definitely deserves publication in SciPost. It easily fulfills several criteria.

Requested changes

Please see the attached report.

Attachment

Recommendation

Publish (surpasses expectations and criteria for this Journal; among top 10%)

  • validity: top
  • significance: high
  • originality: top
  • clarity: high
  • formatting: perfect
  • grammar: perfect

Author:  Žiga Krajnik  on 2025-03-12  [id 5287]

(in reply to Report 2 by Vincent Caudrelier on 2025-01-30)

We thank the referee for their detailed reading of the manuscript and numerous useful comments which have helped to improved the manuscript. We address their points below:

  1. We thank the referee for point out this typo which has been corrected.
  2. As noted by the referee, the literature on discrete and semi-discrete integrable systems is vast and we do not feel qualified to give an overview, however partial. We therefore thank the referee for pointing out these references which have been incorporated into the manuscript.
  3. As noted, the factorization Eq. 1.1 (equivalently Eq. 2.2) is the basic building block for the outlined construction. One key feature that might have been missed by the referee is that while Eq. 2.2 allows for two free parameters, $\lambda$ and $\tau$, we eventually only consider solutions that depend on a single parameter $\tau$, see Eq. 2.3. In other words, we consider solutions that are of difference form: we therefore deal with a single complex spectral parameter $\lambda$, and treat $\tau$ as a fixed lattice parameter. Hopefully this also clarifies several other points raised by the referee. We thank the referee for bringing the reference on entwining Yang-Baxter maps to our attention, which has been incorporated into the manuscript after Eq. 2.3.
  4. We now explicitly mention that the transfer matrices $\mathcal{T}^{(n)}$ are obtained by tracing powers of the monodromy matrix before Eq. 1.5.
  5. As we now state explicitly after Eq. 2.1, by `mutual compatibility' we mean that both Lax operators obey the quadratic Sklyanin bracket with the same $r$-matrix. We also note that $\mathcal{L}$ and $\mathcal{L}^a$ are defined on different, in general non-isomorphic, phase-spaces and therefore their matrix elements trivially Poisson-commute.
  6. In 2.8 we use the refactorization $\mathcal{L}^a_{\lambda^-}(y^\prime)$ $\mathcal{L}_{\lambda^+-\xi}(x^\prime)$ $ = \mathcal{L}_{\lambda^+-\xi} (x)$ $\mathcal{L}^a_{\lambda^-}(y)$. Note that only $\mathcal{L}$ is shifted by the inhomogeneity. As already noted in our response to the third query, we specialize to maps $\psi$ that are of difference form. We can therefore absorb the inhomogeneity $\xi_\ell$ by (locally) shifting the time-step parameter as $\tau \to \tau - \xi_\ell$ yielding $\psi_{\tau - \xi_\ell}$ where $\psi$ is defined as the solution of 2.2. We therefore find it appropriate to say that Eq. 2.7 is resolved by the repeated application of Eq. 2.2. If both Lax operators were shifted by the inhomogeneity all dependence on $\xi_\ell$ could be eliminated by a shift of the spectral parameter $\lambda \to \lambda - \xi_\ell$. The reasons for introducing inhomogeneities in the manuscript was to exhibit the most general form of the construction. In the language of vertex models this means that all parameters on vertical (inhomogeneities) and horizontal (time-step parameters) lines can be chosen independently. The reason for specializing to space-homogeneous models in our applications was to ensure parity of the resulting dynamical systems. While we could also recover parity at the level of inhomogeneity-averaged observables by their appropriate sampling, we have chosen to focus on time-inhomogeneity so as not to unnecessarily obscure the message. We believe the study of space-time inhomogeneous models is an interesting direction for further study.
  7. We thank the referee for catching this typo which has been corrected.
  8. We thank the referee for point this out. We have corrected Eq. 2.21 so that it also holds for $n,m \geq 1$.
  9. As we note after Eq. 2.25, the expansion in Eq. 2.23 merely generates conserved quantities without regard to their locality properties. To generate local conserved quantities one needs to expand the transfer functions around a distinguished (but model-specific) point $\lambda_0$ at which the Lax operators degenerate. In the particular case of the Toda chain local conserved quantities arise by expanding around $\lambda \to \infty$.
  10. While taking the trace of Eq. 2.7 does indeed produce Eq. 2.19 for transfer matrices with spectral parameter $\lambda^+$, one should keep in mind that $\lambda$ is a free parameter. The reported relation 2.19 is obtained by shifting $\lambda \to \lambda - \tau/2$, as we now comment explicitly above Eq. 2.19.
  11. We have reorganized the discussion of the Yang-Baxter relation, and now we start by introducing the intertwiner. As we comment after Eq. 2.27 an additional assumption of uniqueness of factorization is necessary to obtain the Yang-Baxter equation from the equivalence of the products of Lax operators, see also footnote 4. In our reading, this is the content of Proposition 3.1 of reference [3].
  12. To understand the passage from Eq. 2.33 to 2.34 consider Figure 6 (starting on the right-hand side): the physical variables $X$ which are injected at the bottom are subsequently sequentially propagated by the propagators defined by the fixed points $y_2^*$ and $y_1^*$, resulting in the updated physical variables $X'$ exiting on top, achieved by a composition of two left propagators. The pair of fixed points is then mapped by the intertwiner to ${y_1^*}'$ and ${y_2^*}'$, respectively. Note that at this point we have not assumed any particular properties of either ${y_2^*}'$ or ${y_1^*}'$. Consider next the left-hand side of Figure 6. We again input physical variables $X$ at the bottom and $y_2^*$ and $y_1^*$ on the right. Before propagating $X$, the pair of auxiliary variables is first mapped by the intertwiner to ${y_1^*}'$ and ${y_2^*}'$ as defined on the right-hand side. By considering the output auxiliary variables on the right-hand side of Figure 6, we now observe that ${y_1^*}'$ and ${y_2^*}'$ are in fact fixed points of the maps $\Psi_{\tau'}$ and $\Psi_{\tau}$ on the left-hand side, respectively. The propagation of $X$ to $X'$ on the left-hand side is therefore also achieved by the composition of two left propagators. To recognize the significance of the assumption, namely that the intertwiner does not mix fixed points, we emphasize that the propagators are (in case of multiple fixed points) implicitly defined by the choice of fixed point as we note after Eq. 2.18. To obtain the commutation of propagators in Eq. 2.34 from Figure 6 (equivalently Eq. 2.33) it is therefore necessary to assume that the intertwiner maps fixed points of the same type into each other. To give an example, consider the Landau-Lifschitz model where there is a pair of fixed points, an attractive and a repulsive one. The assumption prevents the possibility that a pair of attractive fixed points is mapped to a pair of repulsive fixed points or vice-versa. While we believe that this holds on general grounds, we unfortunately do not see how to explicitly demonstrate this at present. We have expanded the footnote to better clarify the meaning of our assumption.
  13. We now explicitly define $\sigma^0$ after Eq. 4.7 to avoid confusion.
  14. We are grateful to the referee for spotting this mistake. We have rewritten Eqs. 4.8-4.23 with their comments in mind and corrected the mistake in Eq. 4.17. We hope the improved version is now more easily understandable.
  15. One of the motivations for our work was the construction of discretizations that preserve integrability of the underlying continuous-time dynamics. We accordingly felt that that identifying the continuous-time limit naturally belongs to the story, especially since it turned out that the limiting procedure for the Toda lattice is not completely straightforward, involving sending $\tau \to \infty$ and considering the composition of a pair of propagators.
  16. We now use a different font to denote the homology groups.
  17. To aid the identification of Eq. 4.44 with Figure 2 we now give the explicit identification of variables above Eq. 4.44. The variables, $\zeta^a$ stood for the anisotropic stereographic variable of the auxiliary spin, as noted below Eq. 4.69. To avoid any possible confusion over the superscript $a$ we have now changed $\zeta^a \to \zeta^{\rm aux}$. We also specify $X$ in terms of $\zeta$ below Eq. 4.71 and write out the constant $\alpha_\ell$ explicitly in terms of the matrix elements of the Lax operator. 2.We have modified the notation in the discussion on the convergence of fixed point iteration, see Eqs. 4.81-4.89, with the aim of making it more easily understandable. The crucial observation is that, in the limit of a long chain, the average of the sum is given by a uniformly distributed physical spin. The computation is facilitated by noting that the logarithm decouples the integral over the product of factors involving the two auxiliary spins into the sum of two identical integrals which are easily computed.
  18. We thank the referee for catching this typo which has been corrected.

While we are not experts on periodic reductions mentioned by the referee, our understanding is that, while similar to our construction, there are several discernible differences. Periodic reductions apply to multilinear quadrilateral maps defined implicitly by $Q(u, \tilde u, \hat u, \tilde{ \hat u}) = 0$ which is solved by '$3\to 1$' maps $\hat u = F(u, \tilde u, \tilde{ \hat u})$. The initial conditions are given on an initial saw-tooth in the space direction and propagated by alternating application of the '$3 \to 1$' map. Periodic closure of the saw-tooth gives a pair of additional constraints which allows for the elimination of two variables.

By contrast, we deal with '$2\to 2$' maps and specify the physical initial condition along a straight line. Time evolution is then defined at expense of introducing an extra initial auxiliary variable which is sequentially propagated horizontally across the system by sequentially scattering with all of the physical variables. Imposing the periodic closure in the auxiliary direction gives rise to a fixed point condition whose solution permits to eliminate the auxiliary variables from the propagator, thereby defining the time-evolution purely in terms of the physical variables. Our construction therefore effectively eliminates half of the involved variables, unlike the periodic reductions which eliminate only two variables.

Brownian motion of the solitons arises due to a local stochastic rescaling of time generated by multiplicative noise. The increments of a soliton's position is proportional to the time-step. As these are independent and (in the continuous-time limit) normally distributed so are the soliton's increments, a characteristic of Brownian motion. Importantly, it is shown that this does not break integrability of the dynamics since the propagators conserve the transfer matrix for all time-steps, unlike an additive noise in e.g. a Langevin equation which would immediately break all conservation laws.

Author:  Žiga Krajnik  on 2025-03-12  [id 5286]

(in reply to Report 2 by Vincent Caudrelier on 2025-01-30)

We thank the referee for their detailed reading of the manuscript and numerous useful comments which have helped to improved the manuscript. We address their points below:

  1. We thank the referee for point out this typo which has been corrected.
  2. As noted by the referee, the literature on discrete and semi-discrete integrable systems is vast and we do not feel qualified to give an overview, however partial. We therefore thank the referee for pointing out these references which have been incorporated into the manuscript.
  3. As noted, the factorization Eq. 1.1 (equivalently Eq. 2.2) is the basic building block for the outlined construction. One key feature that might have been missed by the referee is that while Eq. 2.2 allows for two free parameters, $\lambda$ and $\tau$, we eventually only consider solutions that depend on a single parameter $\tau$, see Eq. 2.3. In other words, we consider solutions that are of difference form: we therefore deal with a single complex spectral parameter $\lambda$, and treat $\tau$ as a fixed lattice parameter. Hopefully this also clarifies several other points raised by the referee. We thank the referee for bringing the reference on entwining Yang-Baxter maps to our attention, which has been incorporated into the manuscript after Eq. 2.3.
  4. We now explicitly mention that the transfer matrices $\mathcal{T}^{(n)}$ are obtained by tracing powers of the monodromy matrix before Eq. 1.5.
  5. As we now state explicitly after Eq. 2.1, by `mutual compatibility' we mean that both Lax operators obey the quadratic Sklyanin bracket with the same $r$-matrix. We also note that $\mathcal{L}$ and $\mathcal{L}^a$ are defined on different, in general non-isomorphic, phase-spaces and therefore their matrix elements trivially Poisson-commute.
  6. In 2.8 we use the refactorization $\mathcal{L}^a_{\lambda^-}(y^\prime)$ $\mathcal{L}_{\lambda^+-\xi}(x^\prime)$ $ = \mathcal{L}_{\lambda^+-\xi} (x)$ $\mathcal{L}^a_{\lambda^-}(y)$. Note that only $\mathcal{L}$ is shifted by the inhomogeneity. As already noted in our response to the third query, we specialize to maps $\psi$ that are of difference form. We can therefore absorb the inhomogeneity $\xi_\ell$ by (locally) shifting the time-step parameter as $\tau \to \tau - \xi_\ell$ yielding $\psi_{\tau - \xi_\ell}$ where $\psi$ is defined as the solution of 2.2. We therefore find it appropriate to say that Eq. 2.7 is resolved by the repeated application of Eq. 2.2. If both Lax operators were shifted by the inhomogeneity all dependence on $\xi_\ell$ could be eliminated by a shift of the spectral parameter $\lambda \to \lambda - \xi_\ell$. The reasons for introducing inhomogeneities in the manuscript was to exhibit the most general form of the construction. In the language of vertex models this means that all parameters on vertical (inhomogeneities) and horizontal (time-step parameters) lines can be chosen independently. The reason for specializing to space-homogeneous models in our applications was to ensure parity of the resulting dynamical systems. While we could also recover parity at the level of inhomogeneity-averaged observables by their appropriate sampling, we have chosen to focus on time-inhomogeneity so as not to unnecessarily obscure the message. We believe the study of space-time inhomogeneous models is an interesting direction for further study.
  7. We thank the referee for catching this typo which has been corrected.
  8. We thank the referee for point this out. We have corrected Eq. 2.21 so that it also holds for $n,m \geq 1$.
  9. As we note after Eq. 2.25, the expansion in Eq. 2.23 merely generates conserved quantities without regard to their locality properties. To generate local conserved quantities one needs to expand the transfer functions around a distinguished (but model-specific) point $\lambda_0$ at which the Lax operators degenerate. In the particular case of the Toda chain local conserved quantities arise by expanding around $\lambda \to \infty$.
  10. While taking the trace of Eq. 2.7 does indeed produce Eq. 2.19 for transfer matrices with spectral parameter $\lambda^+$, one should keep in mind that $\lambda$ is a free parameter. The reported relation 2.19 is obtained by shifting $\lambda \to \lambda - \tau/2$, as we now comment explicitly above Eq. 2.19.
  11. We have reorganized the discussion of the Yang-Baxter relation, and now we start by introducing the intertwiner. As we comment after Eq. 2.27 an additional assumption of uniqueness of factorization is necessary to obtain the Yang-Baxter equation from the equivalence of the products of Lax operators, see also footnote 4. In our reading, this is the content of Proposition 3.1 of reference [3].
  12. To understand the passage from Eq. 2.33 to 2.34 consider Figure 6 (starting on the right-hand side): the physical variables $X$ which are injected at the bottom are subsequently sequentially propagated by the propagators defined by the fixed points $y_2^*$ and $y_1^*$, resulting in the updated physical variables $X'$ exiting on top, achieved by a composition of two left propagators. The pair of fixed points is then mapped by the intertwiner to ${y_1^*}'$ and ${y_2^*}'$, respectively. Note that at this point we have not assumed any particular properties of either ${y_2^*}'$ or ${y_1^*}'$. Consider next the left-hand side of Figure 6. We again input physical variables $X$ at the bottom and $y_2^*$ and $y_1^*$ on the right. Before propagating $X$, the pair of auxiliary variables is first mapped by the intertwiner to ${y_1^*}'$ and ${y_2^*}'$ as defined on the right-hand side. By considering the output auxiliary variables on the right-hand side of Figure 6, we now observe that ${y_1^*}'$ and ${y_2^*}'$ are in fact fixed points of the maps $\Psi_{\tau'}$ and $\Psi_{\tau}$ on the left-hand side, respectively. The propagation of $X$ to $X'$ on the left-hand side is therefore also achieved by the composition of two left propagators. To recognize the significance of the assumption, namely that the intertwiner does not mix fixed points, we emphasize that the propagators are (in case of multiple fixed points) implicitly defined by the choice of fixed point as we note after Eq. 2.18. To obtain the commutation of propagators in Eq. 2.34 from Figure 6 (equivalently Eq. 2.33) it is therefore necessary to assume that the intertwiner maps fixed points of the same type into each other. To give an example, consider the Landau-Lifschitz model where there is a pair of fixed points, an attractive and a repulsive one. The assumption prevents the possibility that a pair of attractive fixed points is mapped to a pair of repulsive fixed points or vice-versa. While we believe that this holds on general grounds, we unfortunately do not see how to explicitly demonstrate this at present. We have expanded the footnote to better clarify the meaning of our assumption.
  13. We now explicitly define $\sigma^0$ after Eq. 4.7 to avoid confusion.
  14. We are grateful to the referee for spotting this mistake. We have rewritten Eqs. 4.8-4.23 with their comments in mind and corrected the mistake in Eq. 4.17. We hope the improved version is now more easily understandable.
  15. One of the motivations for our work was the construction of discretizations that preserve integrability of the underlying continuous-time dynamics. We accordingly felt that that identifying the continuous-time limit naturally belongs to the story, especially since it turned out that the limiting procedure for the Toda lattice is not completely straightforward, involving sending $\tau \to \infty$ and considering the composition of a pair of propagators.
  16. We now use a different font to denote the homology groups.
  17. To aid the identification of Eq. 4.44 with Figure 2 we now give the explicit identification of variables above Eq. 4.44. The variables, $\zeta^a$ stood for the anisotropic stereographic variable of the auxiliary spin, as noted below Eq. 4.69. To avoid any possible confusion over the superscript $a$ we have now changed $\zeta^a \to \zeta^{\rm aux}$. We also specify $X$ in terms of $\zeta$ below Eq. 4.71 and write out the constant $\alpha_\ell$ explicitly in terms of the matrix elements of the Lax operator. 2.We have modified the notation in the discussion on the convergence of fixed point iteration, see Eqs. 4.81-4.89, with the aim of making it more easily understandable. The crucial observation is that, in the limit of a long chain, the average of the sum is given by a uniformly distributed physical spin. The computation is facilitated by noting that the logarithm decouples the integral over the product of factors involving the two auxiliary spins into the sum of two identical integrals which are easily computed.
  18. We thank the referee for catching this typo which has been corrected.

While we are not experts on periodic reductions mentioned by the referee, our understanding is that, while similar to our construction, there are several discernible differences. Periodic reductions apply to multilinear quadrilateral maps defined implicitly by $Q(u, \tilde u, \hat u, \tilde{ \hat u}) = 0$ which is solved by '$3\to 1$' maps $\hat u = F(u, \tilde u, \tilde{ \hat u})$. The initial conditions are given on an initial saw-tooth in the space direction and propagated by alternating application of the '$3 \to 1$' map. Periodic closure of the saw-tooth gives a pair of additional constraints which allows for the elimination of two variables.

By contrast, we deal with '$2\to 2$' maps and specify the physical initial condition along a straight line. Time evolution is then defined at expense of introducing an extra initial auxiliary variable which is sequentially propagated horizontally across the system by sequentially scattering with all of the physical variables. Imposing the periodic closure in the auxiliary direction gives rise to a fixed point condition whose solution permits to eliminate the auxiliary variables from the propagator, thereby defining the time-evolution purely in terms of the physical variables. Our construction therefore effectively eliminates half of the involved variables, unlike the periodic reductions which eliminate only two variables.

Brownian motion of the solitons arises due to a local stochastic rescaling of time generated by multiplicative noise. The increments of a soliton's position is proportional to the time-step. As these are independent and (in the continuous-time limit) normally distributed so are the soliton's increments, a characteristic of Brownian motion. Importantly, it is shown that this does not break integrability of the dynamics since the propagators conserve the transfer matrix for all time-steps, unlike an additive noise in e.g. a Langevin equation which would immediately break all conservation laws.

---

## Round 1 · Referee Report · Anonymous (Referee 3) · 2025-2-6

Strengths

The manuscript contains novel ideas for the time discretization of the integrable dynamics

Weaknesses

See report

Report

Warnings issued while processing user-supplied markup:

  • Inconsistency: plain/Markdown and reStructuredText syntaxes are mixed. Markdown will be used.
    Add "#coerce:reST" or "#coerce:plain" as the first line of your text to force reStructuredText or no markup.
    You may also contact the helpdesk if the formatting is incorrect and you are unable to edit your text.

The manuscript introduces a new approach to the space and time discretization dynamics of integrable systems. The construction of the chiral propagator seems novel and exciting, even though the connected two-body map has appeared before (e.g. Eq. 4.46 in [11]).

I have conceptual problems with understanding this work and some minor comments.

  1. I wonder what is the connection between the time dynamics generated by the chiral operator and any dynamics generated by the Poisson structure. More specifically, is it possible to interpret Eq. (3.6) as a group version of ${O,H}$ with some H? As the paper is written now, the logic is a bit broken: in (2.22) the commuting flows are introduced, however, the time dynamics is not generated by them!

  2. Perhaps a related question is the canonicity of the chiral operator.
    I am confused by the discussion at the bottom of page 11. The Poisson structure is given by the r-matrix (2.1) and is not necessarily canonical (for example the algebra of spins (4.35)). This suggests that any proof of canonicity should be r-matrix-specific. Surely, locally, in the specific coordinates the Poisson structure is given by (2.35) but do we know the chiral operator in these coordinates?

Additionally, for the Toda chain example, where the Poisson brackets are given by (2.35), the two-body map is NOT canonical!

On top of everything, the following sentence after Eq. (4.15) is very confusing.

" A straightforward calculation shows that the map $\psi_\tau$ is not canonical. Nevertheless, the chiral propagators $\Phi_\tau^\pm$ are canonical, as already established by Sklyanin [23,24,35]."

So, is it actually that the chiral propagators have been introduced by Sklyanin and not by the authors? What part of the proof in Appendix A fails for Toda's two-body map?

  1. From the introduction, it seems that the desired approach to the stochastic integrable systems is to keep integrable dynamics and introduce randomness in the initial conditions. However, the approach of Sec. 5 is different; the dynamics is replaced by the non-integrable averaged propagator (5.5) (as the charges (3.7) cannot be conserved under the linear combination of the chiral propagators unless they are the linear functions). I think discussion of this point should be beneficial for the readers. Additionally, beyond trajectory analysis, examining the time dependence of energy could help to distinguish various regimes.

=======================================================

Minor Comments and Typos

Most minor points and typographical errors have already been addressed in two reports published while this report was in preparation. I will, however, highlight some additional issues and apologize for any redundancies:

  1. What does the sign $\simeq$ in Eq. (2.39) mean? is this that the matrices are proportional to identity? Why then the Toda matrices (4.5) and (4.6) do not satisfy this condition?

  2. The caption for Fig. 3 references Eq. (2.39), but it might be more appropriate to cite Eq. (1.4) instead.

  3. The continuous-time limit for Toda is confusing. Why $\tau\to\infty$? Why the double-step propagator?

  4. In the introduction it is claimed that "An important step forward in this direction has been the development of generalized hydrodynamics, a hydrodynamic theory for integrable systems [8, 9]. However, at present the assumptions of the theory limit its applicability to leading-order dynamics on the ballistic scale. To access dynamics on sub-ballistic scales a recourse to numerical simulations remains a necessity."

Surely, the GHD on the diffusive level has been studied extensively see, for example [1812.00767].

Once the authors have addressed these issues, I will be happy to recommend this manuscript for publication in SciPost.

Recommendation

Ask for major revision

  • validity: ok
  • significance: good
  • originality: good
  • clarity: low
  • formatting: acceptable
  • grammar: -

Author:  Žiga Krajnik  on 2025-03-12  [id 5288]

(in reply to Report 3 on 2025-02-06)

We thank the referee for their reading and evaluation of our work. As noted by the referee, the isotropic two-body spin map is already known. The core idea of this manuscript is, however, to outline how maps of this type can be used to construct integrable circuits with a number of distinct properties. The isotropic spin propagator is merely a widely known and convenient example used to illustrate the general scheme. Their comments are addressed below:

  1. In our understanding, the constructed discrete-time dynamics admits an interpolating Hamiltonian flow generated by some linear combination of commuting Hamiltonians of the continuum theory, as can be readily deduced by the preservation of the classical spectral curve. This explains our introduction of commuting flows. Consequently, the dynamics remain confined to invariant tori of the Hamiltonian counterpart, and the elementary one-step propagator acts simply by discrete shifts (depending on $\tau$) of canonical angle variables. We nonetheless decided to not delve into these aspects (which would require us to introduce the framework of separation of variables) in order to not deviate too much from the main scope of the paper.
  2. The Poisson structure of physical and auxiliary degrees of freedom is indeed prescribed by the $r$-matrix in Eq. (2.1). Concerning canonicity, we believe that there should exist a fairly general (i.e. model-independent) way of establishing this property, but it has so far eluded us. We thus proceed with a case-by-base analysis. The main difference between the examples of Toda and Landau-Lifschitz is that the auxiliary and physical Lax operators are inequivalent in the former case despite sharing the same $r$-matrix. There is no a priori guarantee that $\psi_\tau$, the solution to the local zero-curvature condition Eq. (2.2), represents a canonical map. Indeed, the local Toda propagator Eqs. (4.8-4.11) is an example of a non-cannonical solution. Evidently it then follows that the composite propagators $\Psi^\pm_\tau$ are also not canonical. The physical Toda propagator $\Phi^\pm_\tau$ however becomes canonical after eliminating the auxiliary variable by periodically closing the chain. The solution of the local discrete zero-curvature condition for the Toda lattice is well-known, and to our knowledge was first obtained by Sklyanin who also obtained the equivalent of $\Phi^-_\tau$ and demonstrated its canonicity, as noted in the manuscript, see Refs. [23] and [24]. Our construction shows that the same type of dynamics can be realized for other integrable models with a solution of the discrete zero-curvature condition given as the input. We have also established a number of nontrivial properties of such dynamics in a model-independent manner. Among others, we demonstrate how by alternating left and right propagators (in absence of inhomogeneities) one can remedy the undesired chirality of the elementary one-step propagators. The auxiliary and physical Lax operators of the Landau-Lifschitz are isomorphic. The solutions $\psi_\tau$ of the corresponding discrete zero-curvature conditions are known (see Refs. [11-13]) where it is also demonstrated that they preserve the Poisson structure of Eq. (2.1), from which it immediately shows that they are canonical maps. Note that explicit expressions for Lax operators or propagators in canonical coordinates are unnecessary to obtain this result. Since $\psi_\tau$ is canonical this directly gives that $\Psi_\tau$ is also canonical as we note after Eq. (4.47). It then remains to understand whether eliminating the auxiliary (i.e. horizontal) variable by imposing periodicity spoils the canonicity of the resulting map $\Phi^-_\tau$. The last question is addressed in Appendix A in a model-independent manner. In particular, we show that starting with a canonical map $\Psi_\tau$ and eliminating the auxiliary variables by imposing the periodicty condition Eq. (2.10) necessarily yields a canonical map $\Phi_\tau^-$. Given that our example of the Landau-Lifschitz model produces a canonical $\Psi_\tau$ the proof of Appendix A applies, while the Toda example gives a non-canonical $\Psi_\tau$, hence the reference to Sklyanin's proof.
  3. It appears that there are certain misunderstandings that require some clarification. Firstly, stochasticity enters our construction via the random choice of time-step parameters and thus concerns the time-evolution and not initial conditions. This complies with the conventional use of `stochastic dynamics' in the realm of many-body systems, where randomness enter through the equation of motion. Initial condition are of course random, since we deal with maximal entropy ensembles. But we could just as easily have studied a quench protocol by prescribing some fixed initial configuration. Secondly, the propagator given by equation (5.5) is integrable! Its integrability is guaranteed by the construction outlined in Sections 2 and 3. Crucially, randomness in the time-step parameters does not break integrability. This indeed is the key distinction to integrable brickwork circuits, where picking different $\tau$ along the evolution invariably spoils integrability. In fishnet circuits, exact conservation of the charges, stated in Eq. (3.7), follows from the conservation of the transfer functions. Consequently, if by 'energy' one has in mind one of the conserved quantities as per Eq. (3.7), there are no dynamics.

Minor commnets:

  1. The symbol $\simeq$ indicates proportionality up to a scalar factor as we now state explicitly. As correctly observed by the referee, the Toda Lax operators do not satisfy the inversion relation. Note, however, that the inverse relation is not required for the construction. We made a different statement, namely if Eq. (2.39) is satisfied, then the inverse propagator can be realized by negating the time-step, see Eq. (2.40).
  2. To realize the left propagator in Figure 3 in terms of the right propagator with a reversed direction and a negative time-step we use the inversion relation (2.39) as noted in Eq. (2.42) and the caption, hence its inclusion in the caption of Figure 3.
  3. It is well known that the local conserved quantities of the Toda lattice are obtained by expanding the transfer matrix around $\lambda \to \infty$ (in the standard parametrization of the spectral plane). To retrieve the corresponding continuous-time dynamics it is necessary to let $\tau \to \infty$. As we explain after Eq. (4.33), the doubling of the propagator is necessary to bypass the staggering of signs which obstructs a smooth continuous limit.
  4. We have not expressed ourselves clearly. We are well aware of previous works that have succeeded in deriving the exact transport coefficients (Onsager matrix) governing diffusion-scale dynamics in integrable models. We implicitly had in mind something else: the mechanism for diffusive (Brownian) dynamics outlined in our work is distinct from that treated in the aforementioned works, and presently it is not obvious how to incorporate it within the existing framework of generalized hydrodynamics. The origin of diffusive-scale dynamics in our fishnet circuits is not `diffusion by convention' as in deterministic interacting integrable systems but rather stochasticity in time which we describe by introducing stochastic dressing.

---

## Round 2 · Referee Report · Anonymous (Referee 3) · 2025-3-18

Report

I would like to thank the authors for clarifying most of the issues.
I am still however puzzled about the averaged propagator - for example, given by Eq. (5.5). I understand that the map $\Phi_\tau$ in Eq. (3.5) is indeed, integrable. However, as I have written in my previous comment, I don't understand why the linear combination of these maps with different times is integrable. In fact, what does it even mean to have a linear combination of non-linear maps?

For example, does the map $\alpha \Phi_\tau$ mean that we evolve spins for the time $\tau$ and the obtained result rescale? at each lattice site? What happens with the condition $S_x^2+S_y^2+S_z^2=1$ in this case?
What does it mean to have a map $\Phi_{\tau_1} + \Phi_{\tau_2}$? Do we evolve with time $\tau_1$ and $\tau_2$ and the obtained results just add?

Requested changes

Please clarify the issue in the report. Namely: define the averaged propagator and argue why it is integrable.

Recommendation

Ask for minor revision

  • validity: -
  • significance: -
  • originality: -
  • clarity: -
  • formatting: -
  • grammar: -

Author:  Žiga Krajnik  on 2025-05-07  [id 5460]

(in reply to Report 1 on 2025-03-18)

We thank the referee for their positive reevaluation of our work.

To avoid confusion we now give the stochastic dynamics in Eq. (5.2) and subsequently more explicitly by specifying the averaging of an observable. In practice, the average is obtained by sampling sequences of time steps $\boldsymbol{\tau}$ from $\Omega$, evolving a fixed initial configuration of degrees of freedom with the resulting propagators $\Phi_{\boldsymbol{\tau}}$ and averaging over the results. To avoid confusion we now also refer to this operation as timestep- instead of ensemble-averaging. Timestep averaging should not be confused with averaging over initial conditions (see Eq. (3.12)), but is compatible with it due to linearity of averaging.

---

## Round 2 · Referee Report · Yuan Miao (Referee 1) · 2025-4-12

Strengths

See previous report.

Weaknesses

See previous report.

Report

I have checked the authors' replies to my previous report, which answer most of my (and other referees') comments adequately.

I think that the authors have improved the draft according to the reports and can be published as it is after a small improvement.

A comment on authors' reply of my first comment:
The authors stated that the nomenclature "fishnet circuit" is not related to the "fishnet diagram" of Zamolodchikov et al, therefore it is justified not to mention those literatures on "fishnet diagram" in QFT.
I fully agree with the authors' comment that the two things are not related. However, that is precisely my point that the authors should make a remark in the draft that those two things are not related.
But if the authors insist that they do not want to mention the papers, I would not suggest the change further. At the moment, the authors cited the papers I mentioned at the beginning of the introduction part without any explanation. Either they should follow my suggestion or delete the citation.

Two questions following authors' replies:

First, in the 13th comment, the authors mentioned that the local charges of the Toda lattice are obtained by expansion at $\lambda \to \infty$, thus we need to take the $\tau \to \infty$ limit to obtain the continuous-time limit. In many other instances, the local charges are obtained by expansion at $\lambda \to 0$, is it the case then we should take the analogous $\tau \to 0$ to get the continuous-time limit?

Second, about the authors' 15th comment, does it mean that the fixed-point condition give a similar analogue of the consistency condition when tracing out the transfer matrix in the quantum case? In the quantum case, one can always add a twist to the transfer matrix without destroying the integrability. Would that be related to the fixed points in the classical case?

Requested changes

See above.

Recommendation

Publish (easily meets expectations and criteria for this Journal; among top 50%)

  • validity: top
  • significance: high
  • originality: top
  • clarity: high
  • formatting: perfect
  • grammar: perfect

Author:  Žiga Krajnik  on 2025-05-07  [id 5458]

(in reply to Report 2 by Yuan Miao on 2025-04-12)

We again thank the referee for their positive evaluation of our work. To avoid possible misunderstanding we have added an explanatory comment regarding our use of the name fishnet diagrams in the introduction.

Regarding the two questions:

  1. We believe this to be the case. Provided the local conserved quantities are obtained from the series expansion of the transfer matrix in powers of the spectral parameter, the $\tau \to 0$ limit of the discrete-time brickwork/fishnet dynamics will converge to the continuous-time Hamiltonian dynamics generated by the first (lowest order) conserved quantity in the expansion. An interesting question is how to recover the continuous-time dynamics generated by higher conserved quantities.
  2. So far as we are aware, taking the trace in the auxiliary quantum space does not yield additional consistency conditions. On the other hand we presently also do not see how the fixed-point condition arises in the semi-classical limit of the quantum construction. Adding a twist to the monodromy matrix, which can also be introduced in the classical construction, does not appear to be related to the fixed-point condition.

---

## Round 2 · Referee Report · Vincent Caudrelier (Referee 2) · 2025-4-23

Report

The authors have very thoroughly analysed my comments, made appropriate changes to the manuscript and provided answers that did clarify the points I raised. Thank you for this. There is only one more minor change that I would recommend to avoid some confusion building up from reading this work. Strictly speaking, (2.28) is not the set-theoretic Yang-Baxter equation (which would involve the same map and not two types of maps) but an example of entwining Yang-Baxter equation, as I noted in my previous comments. Other than that, I am happy to recommend publication.

Recommendation

Ask for minor revision

  • validity: -
  • significance: -
  • originality: -
  • clarity: -
  • formatting: -
  • grammar: -

Author:  Žiga Krajnik  on 2025-05-07  [id 5459]

(in reply to Report 3 by Vincent Caudrelier on 2025-04-23)

We again thank the referee for their positive evaluations of the manuscript. The entwining property of the Yang-Baxter equation Eq. (2.28) is now emphasized after Eq. (2.31) to avoid confusion.

---

## Round 2 · Author Response

Revised version.

---

## Round 2 · List of Changes

• Addressed points raised by the referees as specified in the responses to their comments.
  • Added references.
  • Minor corrections and notational clarifications.
  • Fixed grammatical errors.

---

## Round 4 · Referee Report · Anonymous (Referee 3) · 2025-5-22

Report

I am happy with the provided clarification

Recommendation

Publish (meets expectations and criteria for this Journal)

---

## Round 4 · Referee Report · Anonymous (Referee 2) · 2025-5-23

Report

The authors have addressed all my comments and several more from other referees. The original article was already definitely worth publishing modulo these changes so I am happy to recommend publication.

Recommendation

Publish (surpasses expectations and criteria for this Journal; among top 10%)

---

## Round 4 · Referee Report · Yuan Miao (Referee 1) · 2025-6-14

Strengths

See previous report.

Report

The authors have adequately addressed my and other referees' questions/comments. I suggest that the paper to be published as it is.

Recommendation

Publish (easily meets expectations and criteria for this Journal; among top 50%)

---

## Round 4 · Author Response

Revised version.

---

## Round 4 · List of Changes

• Stochastic dynamics given in terms of observables to avoid confusion.
  • Minor clarifications.

---

## Editorial Decision

published